# Systems biology approach reveals a link between mTORC1 and G2/M DNA damage checkpoint recovery

Hui-Ju Hsieh [1], Wei Zhang[1,9], Shu-Hong Lin[2], Wen-Hao Yang [3,4], Jun-Zhong Wang[5], Jianfeng Shen [1], Yiran Zhang[6], Yiling Lu[7], Hua Wang[6], Jane Yu[8], Gordon B. Mills[7] & Guang Peng [1]

Checkpoint recovery, the process that checkpoint-arrested cells with normal DNA repair capacity resume cell cycle progression, is essential for genome stability. However, the signaling network of the process has not been clearly defined. Here, we combine functional proteomics, mathematical modeling, and molecular biology to identify mTORC1, the nutrient signaling integrator, as the determinant for G2/M checkpoint recovery. Inhibition of the mTORC1 pathway delays mitotic entry after DNA damage through KDM4B-mediated regulation of *CCNB1* and *PLK1* transcription. Cells with hyper-mTORC1 activity caused by TSC2 depletion exhibit accelerated G2/M checkpoint recovery. Those *Tsc2*-null cells are sensitive to WEE1 inhibition in vitro and in vivo by driving unscheduled mitotic entry and inducing mitotic catastrophe. These results reveal that mTORC1 functions as a mediator between nutrition availability sensing and cell fate determination after DNA damage, suggesting that checkpoint inhibitors may be used to treat mTORC1-hyperactivated tumors such as those associated with tuberous sclerosis complex.

[1] Department of Clinical Cancer Prevention, The University of Texas MD Anderson Cancer Center, Houston, TX 77030, USA. [2] Department of Epidemiology, The University of Texas MD Anderson Cancer Center, Houston, TX 77030, USA. [3] Department of Molecular and Cellular Oncology, The University of Texas MD Anderson Cancer Center, Houston, TX 77030, USA. [4] Affiliated Cancer Hospital & Institute of Guangzhou Medical University, Guangzhou, Guangdong Province 510095, China. [5] Department of Electrical Engineering, National Kaohsiung University of Applied Sciences, Kaohsiung 80778, Taiwan. [6] Department of Mathematical Sciences, Georgia Southern University, Statesboro, GA 30460, USA. [7] Department of Systems Biology, The University of Texas MD Anderson Cancer Center, Houston, TX 77030, USA. [8] Division of Pulmonary, Critical Care, and Sleep Medicine, University of Cincinnati College of Medicine, Cincinnati, OH 45267, USA. [9]Present address: QIAGEN (Suzhou) Translational Medicine Co., Ltd, Jiangsu Province 215123, China. These authors contributed equally: Hui-Ju Hsieh, Wei Zhang. Correspondence and requests for materials should be addressed to G.P. (email: gpeng@mdanderson.org)

When genomic DNA is damaged, cells activate DNA damage checkpoints to arrest cell cycle at different stages including G1, S, and G2/M, which ensures sufficient time to repair damaged DNA[1]. Given the fundamental role of DNA damage checkpoint in maintaining genomic stability, a complex protein network has been identified for the activation of DNA damage checkpoints including sensor proteins detecting damaged DNA, mediator proteins transducing DNA damage signaling, and effector proteins pausing cell cycle[2]. However, the molecular cues required for cells to recover from DNA damage checkpoints have not been well characterized. These molecular determinants of DNA damage checkpoint recovery are essential for cells to resume the cell cycle and continue their physiology program in order to maintain survival after DNA damage.

Among all DNA damage checkpoints, the G2/M checkpoint restricts mitosis onset in response to a variety of endogenous and exogenous factors including ionizing radiation (IR), DNA replication, and chemotherapy drugs[3]. It serves as the last DNA damage checkpoint before cell division to prevent unrepaired DNA from passing to daughter cells. Thus, in this study, we used a systems biology approach to identify the signaling network required for cells to recover from G2/M DNA damage checkpoint activation.

We report that the mechanistic target of rapamycin complex 1 (mTORC1) pathway is required for mitotic entry after DNA damage. mTOR controls transcription of master mitotic genes such as *CCNB1* (encoding cyclin B1) and *PLK1* (encoding polo-like kinase 1) after DNA damage through regulating histone lysine demethylase 4B (KDM4B). Furthermore, cells with hyper-mTORC1 activity caused by depletion of tuberous sclerosis 2 (TSC2), a negative regulator of mTORC1, exhibit an accelerated G2/M checkpoint recovery. The abrogation of the G2/M checkpoint by WEE1 inhibition can selectively induce mitotic catastrophe and apoptosis in TSC2-depleted cells. In summary, our study uncovers a new function of mTORC1 in regulating DNA damage checkpoint recovery, which creates a therapeutic vulnerability in mTOR-hyperactivated tumors for DNA damage checkpoint inhibitors.

## Results

**Systems biology approach to study G2/M checkpoint recovery.** We first performed the reverse phase protein array (RPPA) in a time series across two p53-proficient cell lines, U2OS and HCT116, which exhibit obvious G2/M checkpoint activation after IR (Fig. 1a)[4]. We treated cells with IR and then arrested cells in the mitotic phase with paclitaxel to ensure that each cell entered mitosis only once. Six time points we chose for RPPA analysis represented the cell cycle kinetics from DNA damage checkpoint activation (a significant reduction of mitotic cells) to recovery (a resurgence of mitotic cells) after IR (Fig. 1b).

To analyze RPPA data (Supplementary Data 1), the expression level of each protein at different time points was normalized by the level at time point 1 (without IR) for each cell line. Simple linear models were constructed to predict the normalized expression of each protein in U2OS by its respective expression in HCT116. Regression equations with a false discovery rate of <0.3 were considered significant, and correlation coefficients ($r$) between 0.7 and 1 were considered to indicate a strong positive relationship. Using this method, we identified 84 proteins whose expression was strongly correlated between U2OS and HCT116 cells (Fig. 1c and Supplementary Data 2). Instead of searching only for molecules with significant fold changes at the protein expression level in the RPPA data, we conducted network analysis using Ingenuity Pathway Analysis (IPA) software to identify the

key determinants within these 84 molecules in the regulation of checkpoint recovery after IR[5]. The top ten canonical pathways by *p*-value included pathways related to cancers and the phosphoinositide 3-kinase (PI3K) signaling network (Supplementary Fig. 1a). The top two networks with the highest scores contained 60 molecules and involved cellular response to IR, including cell death and survival, cellular growth and proliferation, and cancer (Supplementary Table 1). We then merged these two networks and utilized the mathematical tool of network flow, the Ford-Fulkerson algorithm, for analysis (Fig. 1d and Supplementary Data 3)[6]. The interactions between molecules (nodes) are represented by paths with designated flow capacities in the network analysis. The amount of flow that is allowed to go through the network from each source node (upstream regulator) to each sink node (downstream target) provides a measure of the corresponding property of the network. The Ford-Fulkerson algorithm finds the maximum flow, which represents the importance of molecules and interactions as a function of the DNA damage checkpoint recovery network (Supplementary Fig. 1d).

To examine network flow received by *CCNB1* or *CCND1* (encoding cyclin B1 and cyclin D1, respectively, which control cell cycle progression), we chose ten sets of parameters to represent relationships between two molecules in the IPA network (encompassing interaction, direct control, and indirect control) and calculated the number of times each molecule was identified as the upstream regulator (source node) or was identified in the pathways with the maximum flow property in regulating network flow to *CCNB1* or *CCND1*. Our network modeling showed that mechanistic target of rapamycin (*MTOR*), epidermal growth factor receptor (*EGFR*), and androgen receptor (*AR*) were the top three candidates (Fig. 1d, Supplementary Fig. 1b–d, and Supplementary Data 4, 5). Since mTOR belongs to the PI3K-like family, which includes all central kinases in regulating DNA damage response[7], we chose mTOR as our target to study its function in DNA damage checkpoint recovery as predicted by mathematical modeling of RPPA data.

**mTORC1 regulates G2/M checkpoint recovery.** First, we sought to determine whether mTOR deficiency affects DNA damage checkpoint recovery. We partially depleted mTOR using multiple siRNAs or shRNAs in different cell lines to exclude cell type specificity. *MTOR* knockdown impaired cell cycle recovery after IR, but did not significantly affect the activation of the G2/M checkpoint, cell cycle distribution or the accumulation of mitotic cells trapped by paclitaxel (Fig. 2a–e and Supplementary Fig. 2a, b). Protein expression of G2/M cell cycle regulators, such as polo-like kinase 1 (PLK1), cyclin B1, and phosphorylated histone H3 (p-H3), were reduced in *MTOR*-knockdown cells after IR (Fig. 2f and Supplementary Fig. 2c, d). These results suggest that mTOR is required for G2/M checkpoint recovery after IR.

mTOR kinase functions as part of two complexes, mTOR complex 1 (mTORC1) and mTOR complex 2 (mTORC2)[8]. To determine whether mTOR kinase activity is involved in the regulation of G2/M checkpoint recovery, we treated cells with rapamycin (to inhibit mTORC1) or KU0063794 (to inhibit both mTORC1 and mTORC2) and then applied IR. Both mTOR inhibitors not only reduced G2/M checkpoint recovery after IR, but also reduced the percentage of mitotic cells in the absence of DNA damage, which was not induced by *MTOR* knockdown (Fig. 2g and Supplementary Fig. 2e). Thus, we used an inducible mTOR-kinase-dead knock-in cell model, D2338A-cKI, to study the dosage effect of mTOR kinase activity on the G2/M transition (Supplementary Fig. 2f, g). In this model, loss of one copy of mTOR kinase activity (D2338A) did not affect mitotic entry in

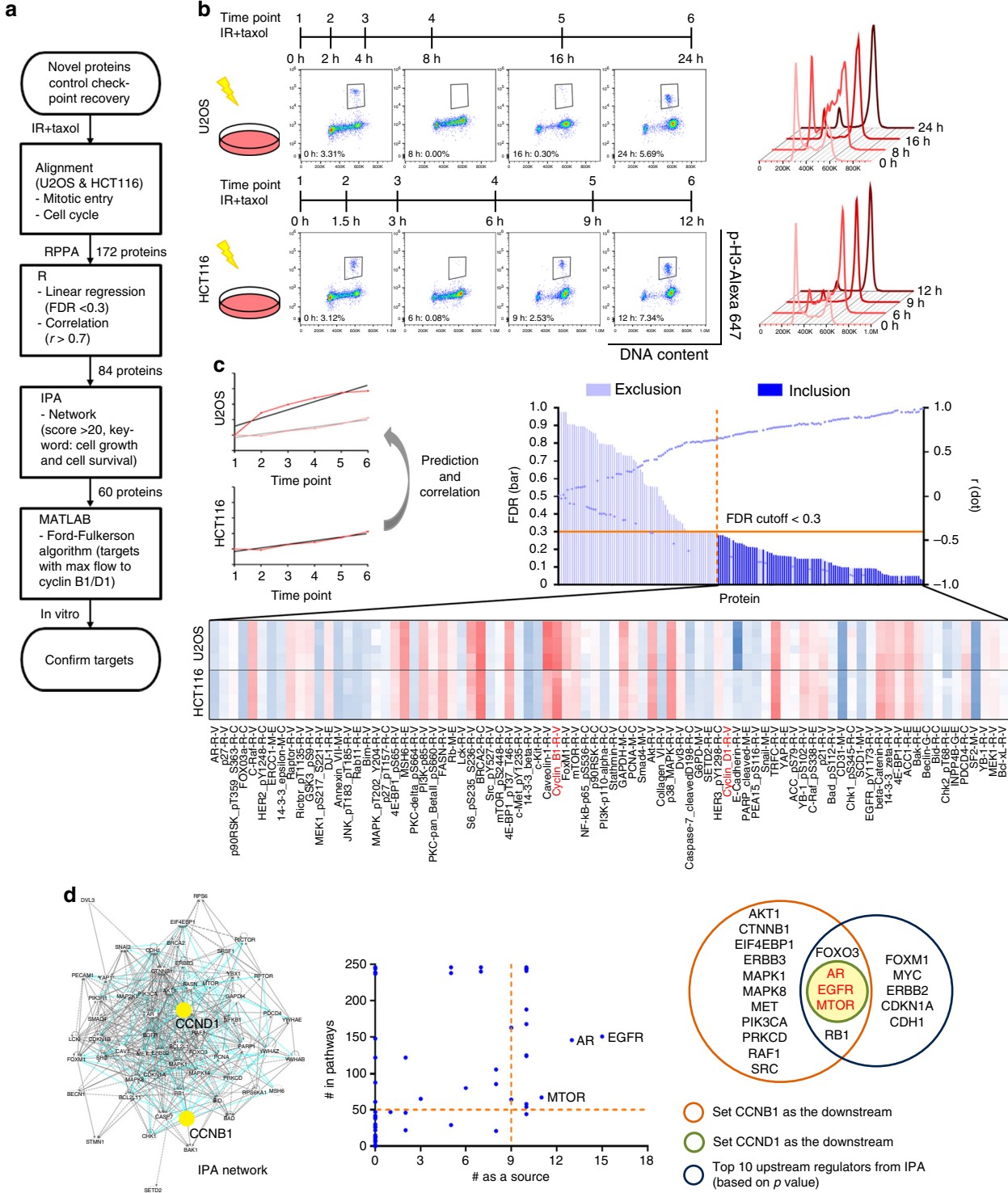

**Fig. 1** mTOR is a candidate for the key molecule regulating G2/M checkpoint recovery. **a** The flow chart demonstrates the process by which we identified candidates involved in DNA damage recovery from RPPA results. **b** RPPA was performed in U2OS cells and HCT116 cells. Cells were irradiated with 7 Gy of IR and then were trapped in the mitotic phase using 2 μM paclitaxel for a period of time. Six time points were chosen on the basis of cell cycle patterns and mitotic entry analysis. The percentage of mitotic cells, defined as p-H3-positive cells, is shown in each representative graph. **c** We used the linear regression slope of each protein in HCT116 cells to predict the same protein expression in U2OS cells and calculate correlations between the two cell lines. Regression equations with a false discovery rate of <0.3 were considered to show a significant linear relationship, and among those proteins, we selected those with a correlation r-value of >0.7 for IPA network analysis. The names in red were two proteins we used as the downstream targets for calculation. **d** We generated the network in IPA. The scatter plot represents the calculation results based on the Ford-Fulkerson algorithm. The potential upstream targets (words in red) came from the comparison between our calculation results and IPA upstream regulator analysis. FDR false discovery rate, taxol paclitaxel

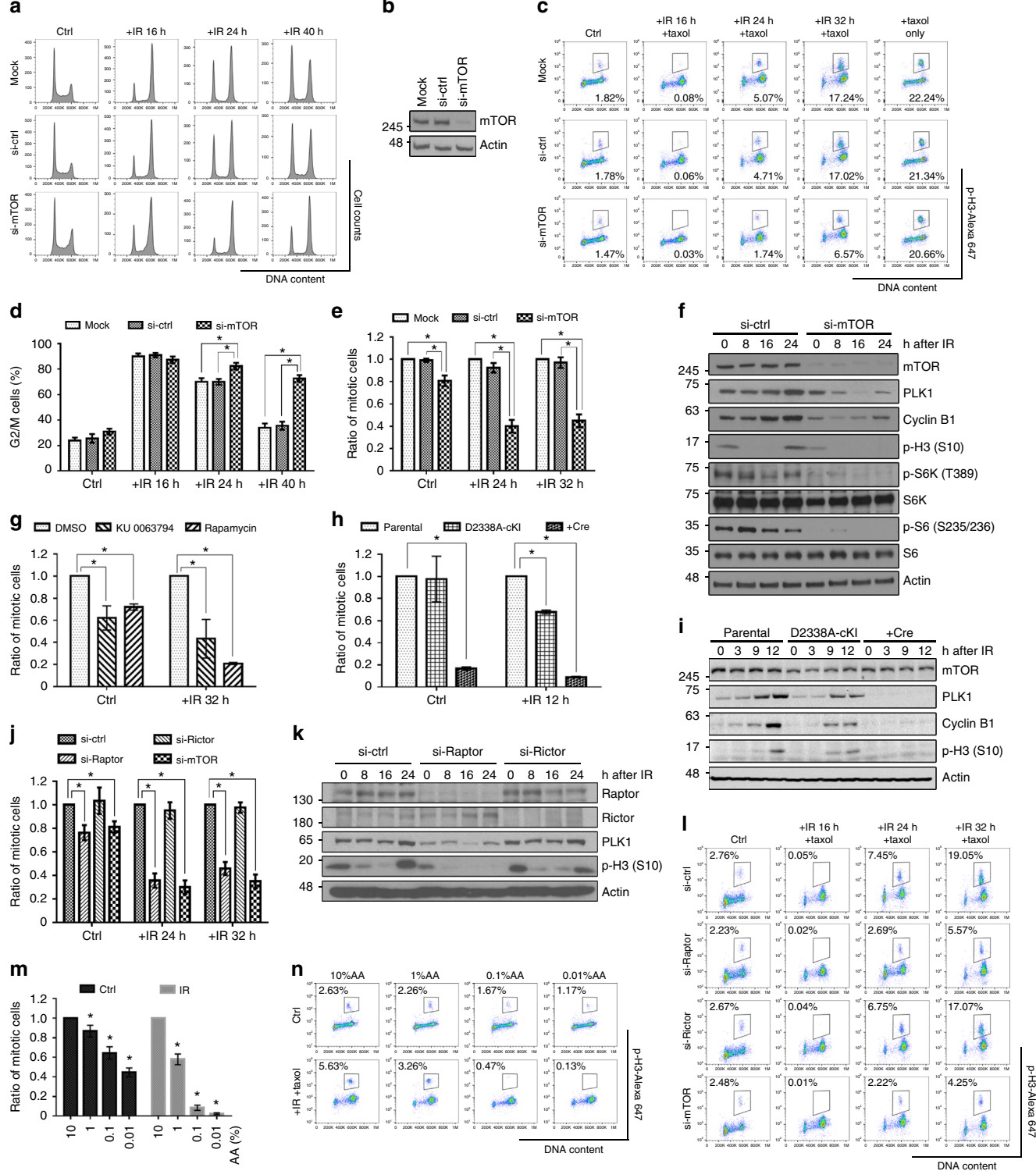

**Fig. 2** mTORC1 regulates G2/M checkpoint recovery. **a**, **d** U2OS cells were collected at different time points after IR (7 Gy) for cell cycle analysis, and the percentages of G2/M cells are presented in **d**. **b** The depletion of mTOR in **a–e** was detected by western blotting. **c**, **e–n** Cells were treated with IR (7 Gy) and 2 μM paclitaxel following different siRNA transfection in **c**, **e**, **f**, **j–l**, different mTOR inhibitor treatments (20 nM rapamycin or 1 μM KU0063794 for 24 h before IR) in **g**, either Ad5-CMV-empty or Ad5-CMV-Cre virus particle infection in **h** and **i**, or different concentrations of amino acids (AA; normal medium with 100% AA, 28 h before IR) in **m** and **n**. Cells were then stained with propidium iodide and p-H3 for mitotic entry analysis . The numbers in representative flow cytometry plots indicate percentages of mitotic cells, which were defined as p-H3-positive cells with 4N DNA contents. Protein samples at different time points were collected for western blotting . Actin was an internal control. Mock: cells incubated with only the transfection reagent; si-ctrl, si-Raptor, si-Rictor, or si-mTOR: cells transfected with non-target control siRNA, raptor, rictor, or mTOR siRNA, respectively; ctrl: control; error bars: mean ± SEM in **d**, **e**, **g**, **j** and mean ± SD in **h**, **m**; $n = 3$ independent experiments; *$p < 0.05$, two-tailed, unpaired Student's $t$-tests

the absence of DNA damage but showed 40% reduction of mitotic entry after IR. However, loss of two copies (+Cre) severely reduced the number of mitotic cells regardless of the presence of DNA damage (Fig. 2h and Supplementary Fig. 2h). Moreover, the expression levels of PLK1, cyclin B1, and p-H3 were positively correlated with mTOR kinase activity (Fig. 2i). These results suggest that partial deficiency in mTOR kinase activity is sufficient to impair DNA damage checkpoint recovery without affecting normal cell cycle transition. To understand the roles of mTOR complexes in DNA damage checkpoint recovery, we depleted raptor and rictor, the specific components of mTORC1 and mTORC2, respectively. We found a recovery defect similar to that due to partial mTOR depletion in *RAPTOR* (encoding raptor)-knockdown cells, but not in *RICTOR* (encoding rictor)-knockdown cells (Fig. 2j–l). These data indicate that mTORC1, but not mTORC2, is required for G2/M DNA damage checkpoint recovery. To confirm the results, we conducted amino acid withdrawal experiments to inhibit mTORC1 activity by causing nutrient deprivation, a physiologically relevant condition of mTOR inhibition[9,10]. When the level of amino acids dropped, cells exhibited a much stronger dose-dependent defect in G2/M checkpoint recovery after IR compared to the cell cycle progression in the absence of DNA damage (Fig. 2m, n). These results suggest that nutrient availability sensing by the mTORC1 pathway plays a more important role in determining cell cycle recovery after DNA damage than it does during normal cell cycle progression.

**mTORC1 regulates *CCNB1* and *PLK1* transcription**. To understand how mTORC1 regulates G2/M checkpoint recovery, a comet assay was performed. We found that both mTOR-deficient cells and control cells showed similar levels of DNA damage, suggesting that defective G2/M checkpoint recovery is not caused by persistent DNA damage in mTOR-deficient cells (Supplementary Fig. 3a). Consistent with the comet assay results, mTOR-deficient cells did not exhibit persistent γ-H2AX activation, a marker for DNA double-strand breaks (DSBs) (Supplementary Fig. 3b)[11]. Moreover, mTOR-deficient cells showed very similar kinetics of phosphorylation of checkpoint kinase 2 (CHK2) and checkpoint kinase 1 (CHK1), two kinases regulating the G2/M checkpoint after IR[12], suggesting that mTOR deficiency did not affect DNA damage checkpoint signaling activation and termination (Supplementary Fig. 3b).

Given the significant reduction of cyclin B1 and PLK1 expression after IR in mTOR-deficient cells, we next determined whether mTOR regulates G2/M DNA damage checkpoint recovery through a transcriptional program. In control cells, both *CCNB1* and *PLK1* mRNA levels decreased in response to IR, allowing G2/M cell cycle arrest. During the checkpoint recovery, both *CCNB1* and *PLK1* mRNA levels increased, which promoted the onset of mitosis after IR. In mTOR-depleted cells, the basal mRNA levels of *CCNB1* and *PLK1* only slightly decreased, with a reduction of less than 20% compared with control cells (Fig. 3a). However, the capacity of mTOR-depleted cells to induce *CCNB1* and *PLK1* expression after IR was significantly impaired, with a 60% reduction in expression levels compared to control cells (Fig. 3a). Raptor-depleted cells produced phenocopies of mTOR-depleted cells that showed a significantly impaired capacity to induce *CCNB1* and *PLK1* mRNA expression after IR, but rictor-depleted cells did not (Fig. 3b and Supplementary Fig. 3c). These results indicate that mTORC1 is required for the transcriptional induction of *CCNB1* and *PLK1*, which permits mitotic onset in G2/M-arrested cells after DNA damage.

Then, we performed dual-luciferase promoter activity assays to demonstrate that mTOR transcriptionally regulates *CCNB1* and

*PLK1*. The luciferase activities driven by *CCNB1* and *PLK1* promoters (*CCNB1*-luciferase and *PLK1*-luciferase) were lower and were reduced even further after IR in mTOR-depleted cells compared to controls (Fig. 3c). In addition, overexpression of wild-type mTOR increased *PLK1*-luciferase activities compared to empty vector or catalytic-dead mTOR overexpression (Fig. 3d and Supplementary Fig. 3d). These results indicate that mTOR kinase activity is involved in transcriptional regulation of key mitotic regulators.

A recent study reported that KDM4B, a key enzyme regulating histone H3 lysine 9 tri-methylation (H3K9me3), activates transcription of Myb-related protein B (BMYB)-regulated genes, including *CCNB1* and *PLK1* [13]. Thus, we tested whether KDM4B is a key molecular node of the transcriptional program driven by mTOR. First we found that KDM4B showed a dynamic change very similar to that of cyclin B1 after DNA damage, and its increased expression preceded the increase of cyclin B1 during checkpoint recovery, suggesting a role of KDM4B in mitotic onset after DNA damage (Fig. 3e). In KDM4B-depleted cells, we observed similar phenotypes to those of mTOR deficiency, including reduced cyclin B1 expression (Fig. 3e and Supplementary Fig. 3e) and the reduced percentage of cells that entered mitosis during G2/M checkpoint recovery (Fig. 3h). The expression of KDM4B itself was reduced in both mTOR-depleted U2OS and HCT116 cells, but mTOR expression was not altered by KDM4B depletion (Fig. 3e, g, and Supplementary Fig. 3e). Furthermore, overexpression of KDM4B in mTOR-depleted cells might rescue cyclin B1 expression (Supplementary Fig. 3f, g), suggesting KDM4B is a downstream effector of mTOR and plays a role in determining checkpoint recovery. The mTORC1 inhibitor rapamycin, but not the protein kinase B (AKT) inhibitor MK2206, also decreased the expression of KDM4B (Fig. 3f and Supplementary Fig. 3h). To further understand how mTOR might regulate KDM4B, we treated cells with MG132, a proteasome inhibitor, and found that mTOR regulated KDM4B protein stability through ubiquitination and a proteasome-mediated pathway (Supplementary Fig. 3i). These results suggest that KDM4B could be a key molecular link between mTOR activity and epigenetic control of the transcription program for G2/M checkpoint recovery.

To test this hypothesis, we performed a chromatin immuno-precipitation (ChIP)-quantitative PCR assay to analyze the enrichment of KDM4B and H3K9me3 at the *CCNB1* promoter region. KDM4B was recruited to the proximal region of the *CCNB1* promoter 4 h after IR treatment, the time point preceding the induction of cyclin B1 expression, and then the recruitment gradually returned to the basal level (Fig. 3i and Supplementary Fig. 3j). However, this enhanced recruitment of KDM4B after IR was impaired in mTOR-deficient cells (Fig. 3i). As a consequence, in control cells, the level of H3K9me3 at the *CCNB1* promoter was remarkably reduced after IR treatment due to the enhanced recruitment of its histone demethylase KDM4B (Fig. 3j). In contrast, in mTOR-deficient cells, the level of H3K9me3 at the *CCNB1* promoter was significantly higher, particularly after IR treatment (Fig. 3j). As a result of the increased H3K9me3 level, the recruitment of a transcription factor of *CCNB1*, BMYB was reduced (Fig. 3k). Collectively, these data show that mTOR deficiency leads to a suppressive chromatin environment owing to impaired KDM4B function, which restricts transcription of mitotic proteins required for checkpoint recovery.

**TSC2-null cells exhibits accelerated G2/M checkpoint recovery**. Tuberous sclerosis complex 2 (TSC2) is a negative regulator of mTOR signaling[14]. Thus, loss of TSC2 leads to hyperactivation of mTORC1 activity. As we expected, *TSC2*-knockdown human

cells showed significantly increased mitosis during the recovery process compared to wild-type cells, indicating accelerated G2/M checkpoint recovery (Fig. 4a, b). Using *Tsc2*-null mouse embryonic fibroblasts (MEFs), we observed that TSC2 deficiency indeed led to accelerated mitotic entry after DNA damage (Fig. 4c). As shown in Fig. 4d, e, *Tsc2*-null MEFs also exhibited increased nuclear KDM4B and increased expression of PLK1 and cyclin B1. In the absence of DNA damage, paclitaxel treatment did not lead to differences in PLK1 or cyclin B1 expression,

suggesting DNA damage-dependent regulation of the G2/M checkpoint. Together, these data demonstrate that mTORC1 plays an important role in regulating G2/M DNA damage checkpoint recovery.

**TSC2-null cells are sensitive to WEE1 inhibition.** Tuberous sclerosis complex is a genetic disease driven by hyperactivation of mTORC1, which is caused by loss-of-function mutations in *TSC1*

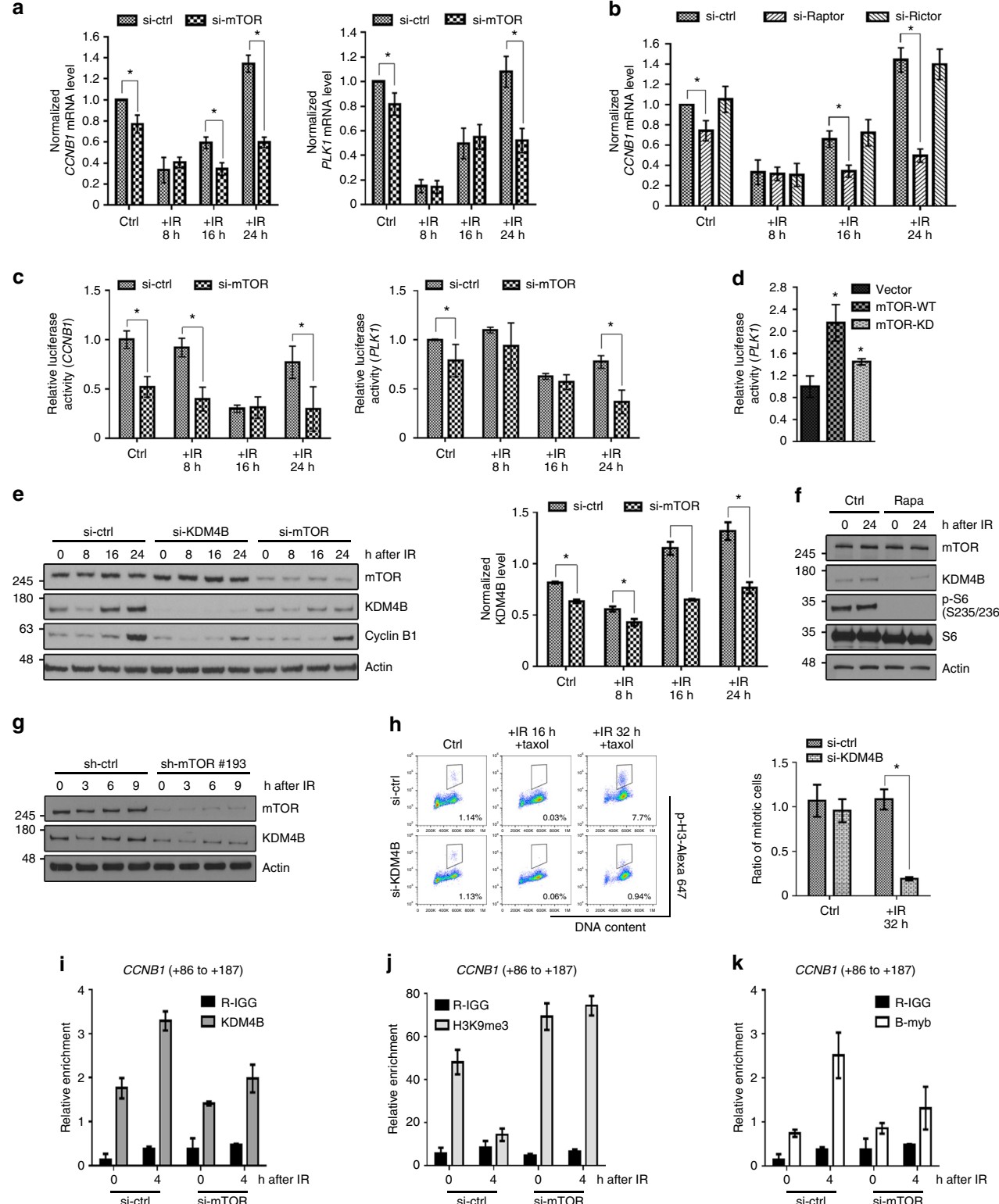

and *TSC2*. Given the accelerated G2/M checkpoint recovery in TSC2-depleted cells, we proposed a synthetic lethal approach that can selectively target *TSC2*-null tumor cells by ablating additional G2/M checkpoint signaling, thus inducing mitotic catastrophe. We treated *Tsc2*-null MEFs with the poly(ADP-ribose) polymerase inhibitor (PARPi) and the WEE1 inhibitor MK1775. PARPi is used as a targeted therapy for treating breast cancer susceptibility gene 1/2 (BRCA1/2)-mutant tumors by causing S-phase-specific DNA damage, which these cells cannot cope with due to their homologous recombination (HR) repair deficiency[15–17]. MK1775 inhibits the catalytic function of WEE1 in maintaining G2 checkpoint arrest, which ensures the completion of DNA repair[18,19]. As shown in Fig. 4f–h and Supplementary Fig. 4a and b, the combination treatment with one potent PARPi, BMN673 or olaparib, and MK1775 at a very low concentration (50 nM) induced a higher number of mitotic catastrophe state such as multipolar mitosis, multinucleated cells, and more apoptotic cells in *Tsc2*-null MEFs compared to wild-type MEFs. *Tsc2*-null MEFs showed increased sensitivity when treated with PARPi alone, likely because of the increased expression of PARP1 in these cells[20]. Nevertheless, the combination of the WEE1 inhibitor and PARPi at low concentrations showed a synergistic effect in selectively targeting *Tsc2*-null MEFs by inducing cell death through mitotic catastrophe.

Previous studies have shown that the effect of WEE1 inhibition is more pronounced in p53-deficient cancers, which have a G1 checkpoint defect and strongly depend on the G2 checkpoint to prevent cell cycle progression[21,22]. *Tsc2*-null MEFs (TSC2−/−) used in our study were p53 deficient. Thus, we tested drug-induced apoptosis in both p53-deficient MEFs (Supplementary Fig. 4c, d) and p53-proficient ELT3 rat cells (Supplementary Fig. 4e, g). Regardless of p53 status, MK1775, BMN673, and the combination treatment induced more apoptosis in *Tsc2*-null cells.

Since our therapeutic strategy targets the molecular consequences of mTORC1 hyperactivation, which is mechanistically different from targeting the hyperactivated mTOR pathway by mTOR inhibitors, we reasoned that this strategy could prove to be a reliable treatment to rapamycin-resistant *TSC2*-null tumor cells. To the end, we developed a rapamycin-resistant *Tsc2*-null cell line (ELT3-V3R) based on *Tsc2*-null ELT3 cells (Eker rat uterine leiomyoma, or ELT3-V3) (Supplementary Fig. 5a–c). The rapamycin-resistant cells exhibited a reduced sensitivity to rapamycin similar to that in *Tsc2*-null cells reconstituted with *Tsc2* expression (ELT3-T3), but they were sensitive to combination treatment (Fig. 4i, Supplementary Fig. 4e–g, and Supplementary Fig. 5a–c). We conducted RPPA to analyze altered molecular signaling in rapamycin-resistant ELT3-V3R cells (Supplementary Fig. 5g, h, and Supplementary Data 6).

Expression of mTOR-related signaling proteins exhibited no remarkable changes in different ELT3 cells. After rapamycin treatment, the expression pattern in ELT3-V3R cells was in general similar to the pattern in its parental ELT3-V3 cells. Notably, rapamycin reduced phosphorylation of ribosomal protein S6 (S6) more significantly in *Tsc2*-null cells (V3 and V3R) compared to *Tsc2*-reconstituted cells (T3). Interestingly, phosphorylation of AKT at serine 473 was significantly higher in ELT3-V3R cells after rapamycin treatment compared to parental ELT3-V3 cells, particularly at a high concentration (20 nM) (Supplementary Fig. 5h). These results were further confirmed by western blotting analysis (Supplementary Fig. 5g) and were consistent with previous publications that phosphorylation of AKT and activation of the mTORC2 complex play an important role in rapamycin resistance[23–26]. Interestingly, *Tsc2*-null cells (ELT3-V3) were more sensitive to rapamycin combined with the WEE1 inhibitor MK1775 compared to rapamycin alone (Supplementary Fig. 5d–f). These data indicate that targeting the G2/M DNA damage checkpoint could be a potential treatment strategy to *TSC2*-null tumors. Next, we conducted an in vivo assay to evaluate the therapeutic effects of targeting the G2/M checkpoint in *TSC2*-null tumors. ELT3 cells were inoculated to the posterior flanks of CB17-SCID mice and the treatment started 5 weeks later when tumors developed. Body weight was monitored to evaluate the toxicity of treatment throughout the experiment, and no significant body weight loss was observed (Fig. 5a). The WEE1 inhibitor MK1775 alone at a clinically relevant dosage (60 mg kg⁻¹) already showed its inhibitory effect on *Tsc2*-null tumor growth in vivo (Fig. 5a–e). Apoptosis was remarkably induced in MK1775-treated tumors compared to vehicle control tumors (Fig. 5f). These results raise the possibility of using MK1775 as an agent to target *TSC2*-null tumor cells. Since we did not observe significant effects with 50 nM MK1775 treatment, a very low concentration optimized to act in synergy with PARPi without altering cell death on its own (Fig. 4f–i, Supplementary Figs. 4a, d–f, and 5e–f), we treated *Tsc2*-null MEF cells in vitro with different dosages of MK1775. As shown in Fig. 5g, compared with *Tsc2*-null cells reconstituted with *Tsc2* expression, *Tsc2*-null cells were more sensitive to MK1775 at concentrations physiologically relevant to in vivo study (100–500 nM)[19]. This result was confirmed by a 3D culture system, where *Tsc2*-null cells showed increased sensitivity to MK1775 compared to wild-type MEF cells (Fig. 5h). The increased sensitivity was not dependent on the WEE1 expression level, as TSC2 deficiency did not change WEE1 expression (Fig. 5i). Cell cycle analysis showed that MK1775 treatment arrested cell cycle progression of wild-type MEF cells, likely because of DNA damage induced by MK1775. However, *Tsc2*-null cells exhibited an increased distribution of cells in the

**Fig. 3** KDM4B links mTORC1 to positive transcriptional control of *CCNB1* and *PLK1*. **a**, **b** U2OS cells transfected with siRNAs were collected at different time points after IR (7 Gy) and 2 μM paclitaxel treatment. *CCNB1* and *PLK1* mRNA levels were measured by quantitative reverse transcriptase-PCR and normalized to actin. **c** The dual-luciferase reporter assay was conducted in U2OS cells transfected with siRNA. The value of firefly luciferase driven by the *CCNB1* or *PLK1* promoter was normalized to the Renilla-luciferase value. **d** We used the dual-luciferase reporter assay in U2OS cells expressing the control vector, wild-type mTOR (mTOR-WT), or kinase-dead mTOR (mTOR-KD) constructs. **e** U2OS cells transfected with siRNAs were collected at different time points after IR and paclitaxel treatment for western blotting. The bar graph shows the KDM4B protein expression level normalized to actin in each group. **f** U2OS cells treated with or without 20 nM rapamycin were exposed to IR and paclitaxel. **g** We depleted mTOR by an individual shRNA (sh-mTOR #193) in HCT116 cells and treated cells with IR (7 Gy) plus 2 μM paclitaxel. **h** U2OS cells transfected with control or KDM4B siRNAs were exposed to IR (7 Gy) plus 2 μM paclitaxel and were stained with propidium iodide and p-H3 for mitotic entry analysis. The numbers in the representative flow cytometry plots indicate the percentages of p-H3-positive stained cells. **i–k** U2OS cells treated with IR (7 Gy) were collected for ChIP analysis using anti-KDM4B, anti-H3K9me3, or anti-BMYB antibody. The immunoprecipitated DNA fragments were amplified with the primer to the *CCNB1* transcription regulation region. si-ctrl, si-Raptor, si-Rictor, si-KDM4B, and si-mTOR: cells transfected with non-target control siRNA, raptor, rictor, KDM4B, and mTOR siRNA, respectively; ctrl, control; rapa, rapamycin; taxol, paclitaxel; R-IGG: normal rabbit IgG; error bars: mean ± SD; *n* = 3 independent experiments; *p < 0.05, two-tailed, unpaired Student's *t*-tests

G2/M phase and also an increased number of mitotic cells 24 h after MK1775 treatment compared to wild-type cells (Fig. 5j, k). These data suggest that *Tsc2*-null cells have an increased dependence on WEE1 to maintain the G2/M checkpoint. As a consequence of lacking proper G2/M arrest, DNA damage induced by MK1775 treatment could not be efficiently repaired in *Tsc2*-null cells even though *Tsc2*-null cells initially had a lower level of DSBs indicated by γ-H2AX compared to wild-type cells

(Fig. 5l). Together, the WEE1 inhibitor can be used for *TSC2*-null tumor treatment by accelerating release from the G2/M DNA damage checkpoint dampened by hyperactivation of mTORC1.

## Discussion
Checkpoint recovery does not simply reverse the checkpoint activation process. To identify key regulators of G2/M DNA

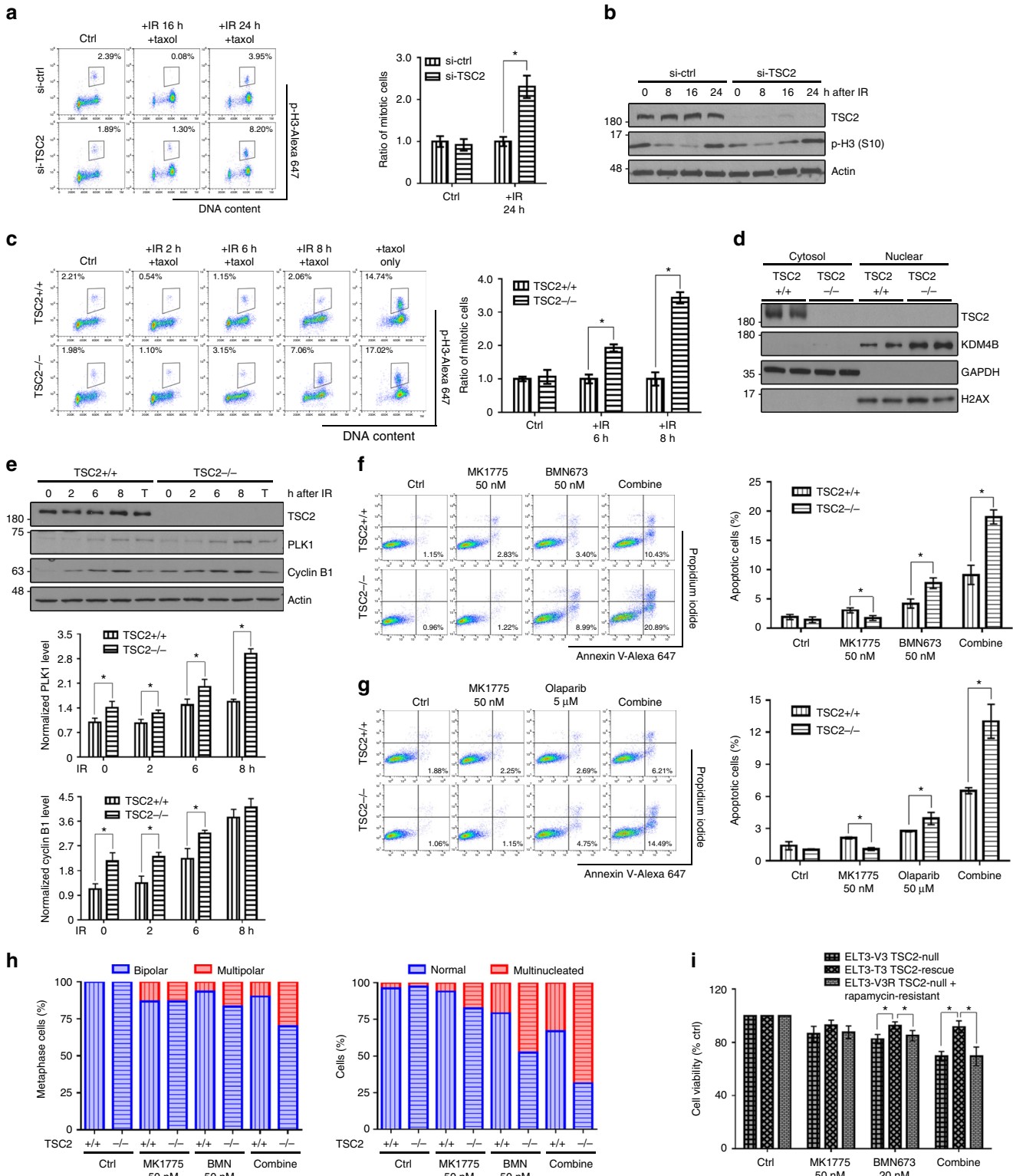

damage checkpoint recovery, we analyzed proteomic data through a systems biology approach and discovered the function of mTOR signaling in regulating the KDM4B-mediated transcriptional program, which is required for recovery of the G2/M DNA damage checkpoint. This mechanism provides a new avenue for targeting abnormal G2/M checkpoint recovery in tumor cells with mTORC1 hyperactivation through the use of G2/M checkpoint inhibitors to induce mitotic catastrophe and cell death.

During checkpoint recovery, RPPA data showed that a variety of proteins alter their expression. In the context of the signaling network, the impact of a protein on the overall biological function of this network is determined not only by the fold change of its expression but also by the structure of the network and the connectivity of this protein with other proteins in this network. Thus, instead of selecting the candidate with the most significant change in its protein expression, we used network flow to identifying the key regulatory components. The process of analysis is similar to that for solving maximum flow problems in mathematics. It involves finding a maximum flow (signaling cascade) through a flow network (signaling network) with a source (upstream molecule) and a sink (downstream molecule). We chose the Ford-Fulkerson algorithm, which repeatedly finds augmenting paths in the network and improves the total amount of flow until the flow reaches the maximum. This approach revealed that mTOR is a central regulator in the signaling network regulating G2/M checkpoint recovery, even though its expression change was not the biggest among all candidates indicated by RPPA data. The validation of this prediction in this study demonstrated that our mathematical modeling algorithm may provide a new tool to analyze high-throughput protein expression data at the network level and to identify key regulatory components in a given protein network.

mTOR regulates cell growth and cell division in response to growth factors, energy status, nutrients, and stress[8,10]. mTORC1 is the main complex that couples environmental cues, especially the availability of amino acids, to cell cycle progression[9,27–30]. Thus, it is not surprising that mTOR deficiency can alter mitotic entry in normal cell cycle progression. However, mTOR-deficient cells show a major defect in recovering from the G2/M checkpoint after DNA damage. The defect in G2/M checkpoint recovery is not caused by persistent checkpoint signaling mediated by ataxia-telangiectasia-mutated (ATM)-CHK2 and ataxia telangiectasia and Rad3-related (ATR)-CHK1 or by persistent DNA damage due to inefficient repair in cells, although inhibition of mTORC1 by rapamycin has been suggested to reduce HR repair[31]. Our study reveals that the G2/M checkpoint recovery defect caused by mTOR deficiency is through mTORC1 transcriptional regulation of mitotic proteins cyclin B1 and PLK1, and

the defect is in a dose-dependent manner. In the knock-in mTOR-kinase-dead model, the heterozygous mTOR kinase mutant showed reduced recovery capacity while the homozygous mTOR kinase mutant showed a more severe defect. Amino acid starvation also showed a dose-dependent response in terms of G2/M checkpoint recovery. At the same time, partial loss of mTOR does not arrest the cell cycle at G1 or reduce HR repair as rapamycin does. These data suggest that partial deficiency in mTOR activity is sufficient to block cells from G2/M checkpoint recovery without changing normal cell cycle transition or the capacity of DNA repair.

mTOR controls the expression of KDM4B, a histone demethylase that selectively demethylates H3K9me2/me3 to H3K9me1/me2, and further regulates the KDM4B-mediated transcriptional program of cyclin B1 and PLK1 specifically during G2/M checkpoint recovery. It is of future interest to determine how mTOR regulates KDM4B expression and whether KDM4B is a substrate of mTOR, which might directly affect its ubiquitination and protein degradation. We also observed that KDM4B expression increases during G2/M checkpoint recovery after DNA damage. A positive feedback loop through protein stability or transcription might be involved in regulating KDM4B after DNA damage; such a loop exists for many important mitotic regulators, including PLK1 and cyclin B1[32,33].

Tuberous sclerosis complex is a genetic disease caused by either TSC1 or TSC2 mutation, leading to mTORC1 hyperactivation, and most patients with this disease have TSC2 mutations[34,35]. Although patients with tuberous sclerosis complex usually have benign tumors, these tumors can be life-threatening and difficult to be removed by surgery due to their locations[36,37]. Currently, inhibition of mTOR activity by rapalogs is the only available therapeutic strategy to control tumor growth in these patients[14,36–38]. However, not all patients respond to rapalogs or can tolerate adverse effects of rapalogs, such as immune suppression[39–41]. We demonstrated a strategy to treat TSC2-null tumors using a WEE1 inhibitor via targeting the G2/M checkpoint abnormality, a molecular consequence of mTOR hyperactivation. WEE1 phosphorylates CDK1 and inhibits G2/M transition[18]. Inhibition of WEE1 by MK1775 abrogates G2/M arrest, resulting in premature mitotic entry and mitotic catastrophe[21,22]. In our study, MK1775 induced DNA damage in both wild-type and Tsc2-null cells, likely because of the perturbation of DNA replication due to WEE1 inhibition. It is noteworthy that Tsc2-null cells showed less but persisted DSB formation, as indicated by γ-H2AX formation, compared to wild-type cells. We suspect that TSC2 has a function in regulating DSB formation during DNA replication, which will be further elucidated in our future studies. Together, in TSC2-deficient cells with constitutive high mTORC1 activity, the WEE1 inhibitor

**Fig. 4** High mTORC1 activity facilitates recovery from G2/M checkpoint and promotes sensitivity to WEE1 and PARP inhibition. **a, b** We transfected either non-target control siRNA (si-ctrl) or TSC2 siRNA pool (si-TSC2) into U2OS cells and treated the cells with IR (7 Gy) plus 2 μM paclitaxel. The mitotic percentage (p-H3-positive cells) is shown in the plots. p-H3 protein expression was also detected. **c, e** MEFs were irradiated immediately followed by paclitaxel treatment. Cells treated with paclitaxel alone (the "+ taxol only" and the "T" groups) were collected after 8 h of treatment. For mitotic entry analysis, the numbers in the plots indicate the percentages of p-H3-positive cells. **d** MEFs treated with IR plus paclitaxel for 2 h were separated into nuclear and non-nuclear fractions by the Dounce homogenizer. **f, g** MEFs were incubated with 50 nM MK1775 or/and one PARP inhibitor, 50 nM BMN673 or 5 μM olaparib, for 48 h and were stained with annexin V and propidium iodide. Apoptotic cells were defined as annexin V-positive cells. The percentages of apoptotic cells are shown in representative plots. **h** MEFs were treated with 50 nM MK1775 or/and 50 nM BMN673 for 36 h and were stained with α-tubulin and DAPI. The numbers of centrosomes and nuclei per cell were calculated. **i** We treated ELT3 cells with 50 nM MK1775 or/and 20 nM BMN673 for 4 days. Cell viability was measured by MTT assay. TSC2+/+: Tsc2+/+, Tsc2 wild-type; TSC2−/−: Tsc2−/−, Tsc2 null; ELT3-V3: Tsc2-null Eker rat uterine leiomyoma cells with the control vector; ELT3-T3: Tsc2-null ELT3 cells reexpressing Tsc2; ETL3-V3R: rapamycin-resistant ELT3-V3 cells; ctrl: control; error bars: mean ± SEM in **a**, **c** and mean ± SD in **e–g, i**; n = 3 independent experiments; *p < 0.05, two-tailed, unpaired Student's t-tests

accelerates G2/M transition, induces DNA damage, and increases mitotic catastrophe. The mTORC1 pathway intersects with key mitogenic signals driving tumor development, such as mutations of PI3K, EGFR, K-ras, or loss of phosphatase and tensin homolog (PTEN), which can also lead to hyperactivation of mTORC1 in numerous human cancers[42]. Thus, our study provides a rationale for using G2/M checkpoint inhibitors such as WEE1 inhibitors to target not only TSC2-deficient cells but cancer cells with hyperactivation of the mTORC1 pathway[43].

We further combined MK1775 with PARP inhibitors (BMN673 or olaparib), which can cause more significant S-phase DNA damage, and the dual treatment worked better than any single agent in TSC2-depleted cells. The concentration of MK1775 (50 nM) used in the combination treatment was much lower than that used in monotherapy (200 nM). These data suggest that the combination approach may lower the drug concentrations required to be effective and thus reduce potential toxicity. More interestingly, in rapamycin-resistant

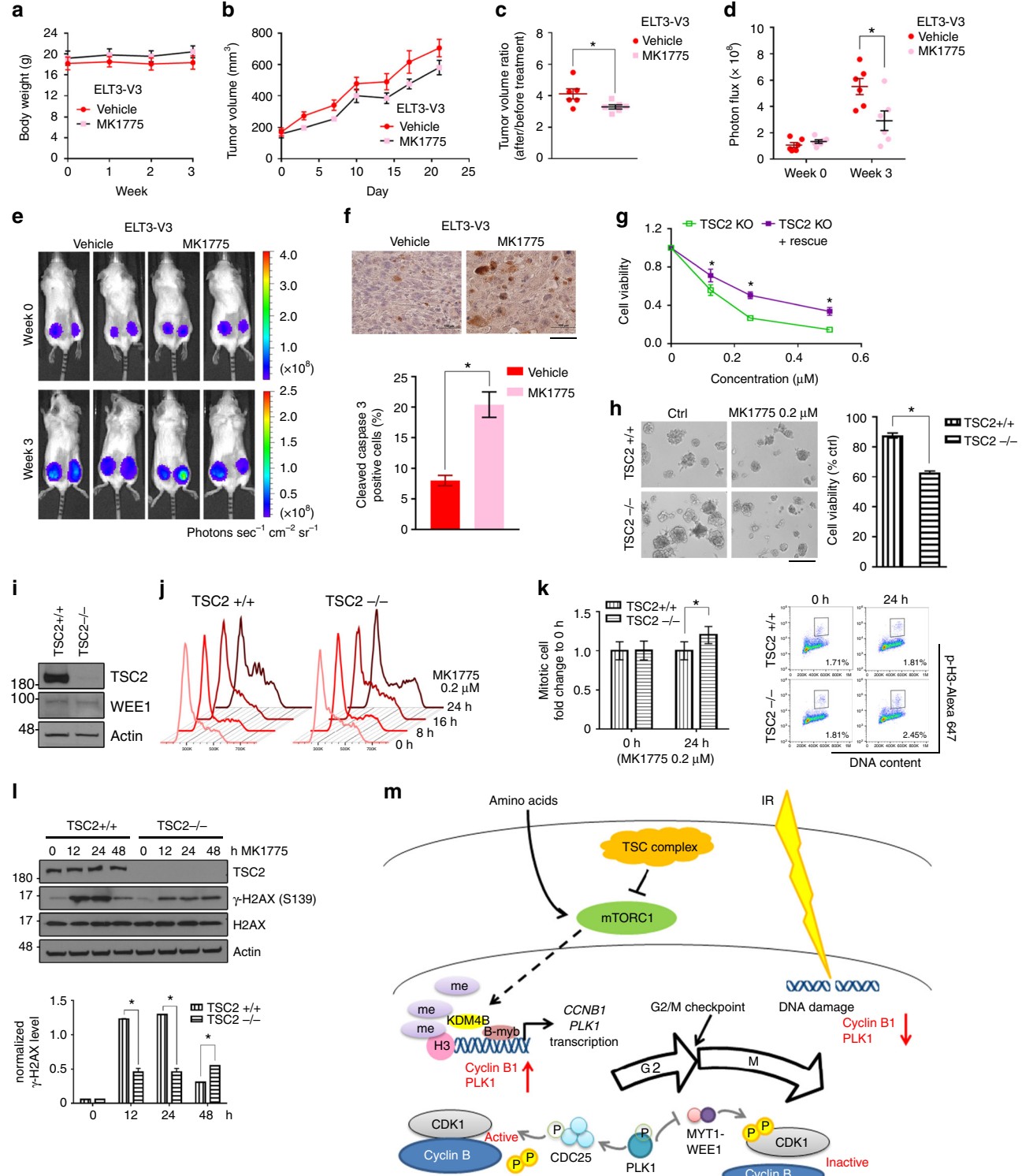

TSC2-depleted cells, the sensitivities to MK1775 and the combination of MK1775 and rapamycin were similar to those in parental TSC2-depleted cells, suggesting that this new strategy might be used to overcome rapamycin resistance in TSC patients due to its novel mechanism of action. Notably, it has been reported that PLK1 interacts with multiple components of the mTOR pathway[44–47]. It remains to be determined whether DNA damage regulates the interaction between PLK1 and components of the mTOR pathway. DNA damage can induce a variety of post-translational modifications such as phosphorylation. It is very likely that these DNA damage-induced protein modifications may alter PLK1–mTOR interactions, which may also contribute to response of *TSC*-null cells to MK1775 and/or BMN673.

In conclusion, we used a systems biology approach to analyze proteomic data during G2/M DNA damage checkpoint recovery, and discovered that mTOR signaling plays an essential role in regulating a transcriptional program required for G2/M checkpoint recovery (Fig. 5m) . This mechanism provides a new therapeutic approach for mTOR-hyperactive tumors using DNA damage checkpoint inhibitors such as WEE1 inhibitors, and this approach has potential to be translated into clinical studies.

## Methods

**Cell culture, IR, and chemicals**. ELT3-V3/T3 cells were obtained from Dr. Jane Yu (University of Cincinnati)[48]. *Tsc2* knockout and reconstituted MEFs were obtained from Dr. David Kwiatkowski (Harvard University)[49]. HCT116 cells (ATCC) and MEFs were cultured in high-glucose Dulbecco modified Eagle's medium. U2OS cells (ATCC) were cultured in McCoy's 5A medium. mTOR kinase-dead conditional knock-in HCT116 cells (D2338A-cKI) were cultured in Roswell Park Memorial Institute 1640 medium with 2 mM L-glutamine and 25 mM sodium bicarbonate as suggested by the manufacturer (Horizon Discovery). All mediums contained 10% fetal bovine serum (Thermo Fisher Scientific) and 1× penicillin–streptomycin (Sigma-Aldrich). All cells were free of mycoplasma contamination and were grown in a humid incubator with 5% $CO_2$ at 37 °C.

To make amino acid-positive medium, we added 1× minimum essential medium amino acids, 1× minimum essential medium non-essential amino acids, and 1× L-glutamine (Invitrogen) into amino acid-free Roswell Park Memorial Institute medium with 10% fetal bovine serum. We then mixed amino acid-free medium (0% AA) with amino acid-positive medium (100% AA) to make 0.01–10% amino acid medium.

To generate the rapamycin-resistant cell line ELT3-V3R, we cultured *Tsc2*-null ELT3-V3 cells in a very low density (500 cells per well in the six-well plate) with 10 nM rapamycin for one week. We then sub-cultured the remaining cells as a pool and treated cells with rapamycin from low concentration to high concentration (increased concentration every passage from 2.5 nM, 5 nM, 10nM to 20 nM, around every 3 or 4 days). Rapamycin was added immediately after cells attached to the plates. The rapamycin-resistant *Tsc2*-null (ELT3-V3R) cells were then maintained in regular medium with 20 nM rapamycin.

We irradiated MEFs with 15 Gy and the rest of the cells with 7 Gy using high-voltage X-ray tubes (RS 2000 Biological Research Irradiator; Rad Source Technologies). We treated cells with 2 μM paclitaxel (Sigma-Aldrich) after IR to arrest cells in the mitotic phase (HCT116: 2 h after IR; U2OS: 6 h after IR; MEFs: immediately after IR). We used 20 nM rapamycin (Sigma-Aldrich) or 1 μM KU0063794 (Selleck Chemicals) to inhibit mTOR kinase activity, 0.1 μM MK2206 to inhibit AKT, MK1775 (Selleck Chemicals) to inhibit WEE1 kinase activity, and the PARP inhibitors BMN673 and olaparib (Selleck Chemicals) for combination therapy.

**Plasmids, shRNAs and siRNAs**. Myc-tagged mTOR-wild-type plasmid, Myc-tagged mTOR-kinase-dead plasmid, control shRNA, and mTOR shRNA #193 were provided by Dr. Dos Sarbassov. KDM4B-wild-type plasmid was purchased from Addgene. We purchased mTOR and KDM4B SMARTpool siRNA, and control siRNA from GE Dharmacon. The individual siRNAs, including si-mTOR (#1: 5′-G GCCAUAGCUAGCCUCAUA-3′ and #2: 5′-CAAAGGACUUCGCCCAUAA-3′), si-raptor (5′-GGACAACGGCCACAAGUAC-3′) and si-rictor (5′-ACUUGUGAA GAAUCGUAUC-3′), were synthesized by Sigma-Aldrich.

**RPPA and computer core availability**. We mainly followed the lysate preparation protocol provided by the Functional Proteomics RPPA Core Facility at The University of Texas MD Anderson Cancer Center. U2OS and HCT116 cells were seeded in six-well plates, and the final cell amounts in each sample fit the minimum requirement of RPPA. Cells were irradiated with 7 Gy and then were incubated with 2 μM paclitaxel at the indicated time points. We lysed the cells with lysis buffer (1% Triton X-100, 50 mM HEPES pH 7.4, 150 mM NaCl, 1.5 mM $MgCl_2$, 1 mM EGTA, 100 mM NaF, 10 mM sodium pyrophosphate, 1 mM $Na_3VO_4$, 10% glycerol, protease inhibitors and phosphatase inhibitors). Protein concentration in the supernatant was determined and was adjusted to 1–1.5 μg μl$^{-1}$. All antibodies (172 antibodies) used in our study were derived from a standard set of antibodies validated and used in MD Anderson Cancer Center Functional Proteomics Core Facility (https://www.mdanderson.org/research/research-resources/core-facilities/functional-proteomics-rppa-core.html). Basically, the protein levels derived from RPPA were correlated with the density of the single immunoblot band (a Pearson correlation coefficient $R \geq 0.7$) and the reproducibility of RPPA was judged by intra- and inter-slide variations (coefficient of variation <15%)[50]. The RPPA data were normalized by the RPPA Core Facility and were further analyzed with the statistical programming language R, Ingenuity Pathway Analysis (IPA; QIAGEN)[51], and MATLAB (MathWorks). The file "RPPA.R" was written for regression and correlation analysis (Supplementary Software). The files "bfs_augmentpath.m" and "show_ff_max_flow.m" were written on the basis of the Ford-Fulkerson algorithm and breadth-first search, but we do not claim authorship of them (Supplementary Fig. 1c, Supplementary Software). The files "output_data.m" and "save_data_build.m" were used to export and save data to Microsoft Excel (Supplementary Software). We used the file "main.m" to control all other MATLAB files and to generate the array for further calculation (Supplementary Fig. 1b and Supplementary Software). All MATLAB files can be opened in the free program Notepad++ (https://notepad-plus-plus.org/) in the Microsoft Windows environment.

To generate the RPPA heat map from ELT3 cells, the antibody staining signals were log2 transformed, centered, and scaled to each antibody. Hierarchical clustering was performed on transformed data using Euclidean distance and complete agglomeration.

**Cell cycle analysis and mitotic entry by flow cytometry**. For cell cycle analysis, cells were fixed in 70% ethanol for at least 2 h and then incubated in propidium iodide solution (10 μg ml$^{-1}$ propidium iodide and 5 μg ml$^{-1}$ RNase A in phosphate-buffered saline (PBS) with Tween-20). To calculate mitotic cell percentages, cells were fixed in 70% ethanol, incubated in permeabilization buffer

**Fig. 5** TSC2 is a potential therapeutic target for WEE1 inhibition. **a–f** ELT3-V3-luciferase cells were injected into mice. Five weeks later, mice were treated with the vehicle or 60 mg kg$^{-1}$ MK1775 three times a week. **a** Mice body weights were monitored weekly for potential drug toxicity. **b, c** Tumor volumes using the formula (length × width$^2$)/2, and **d, e** bioluminescence levels were monitored regularly. **f** Tissue sections were stained with cleaved-caspase 3 and hematoxylin and the graph represents the percentage of cleaved-caspase 3-positive cells in each group. (The scale bar is 100 μm.) **g** MEFs were treated with different concentrations of MK1775 in 96-well plates for 4 days, and cell viability was measured by MTT assay. MK1775 sensitivity was presented as the ratio to the untreated group in each cell line. **h** MEFs were embedded in Matrigel (day 0) and treated with 0.2 μM MK1775 from day 3. The medium with or without MK1775 was changed every 3 days. The representative photos were taken on day 10, and the graph shows the ratio of treated to untreated colonies in each cell line. (The scale bar is 200 μm.) **i** WEE1 protein expression levels were checked in MEFs. **j, l** MEFs treated with 0.2 μM MK1775 were collected at different time points for cell cycle analysis and western blotting. The graph in **l** indicates expression of γ-H2AX normalized to actin in each group. **k** MEFs treated with 0.2 μM MK1775 for 24 h were collected for mitotic entry analysis. Percentages of mitotic cells are shown in representative plots. **m** The schematic summarizes how mTOR controls G2/M checkpoint recovery. The nutrient sensor mTORC1 facilitates G2/M DNA damage checkpoint recovery through an increase in KDM4B-mediated regulation of *CCNB1* and *PLK1* transcription. TSC2+/+: *Tsc2*$^{+/+}$, *Tsc2* wild type; TSC2−/− and TSC2 KO: *Tsc2*$^{-/-}$, *Tsc2* null; TSC2 KO+ rescue: *Tsc2*-null cells with reconstitutive *Tsc2*; ELT3-V3: *Tsc2*-null Eker rat uterine leiomyoma cells with the control vector; ELT3-T3: *Tsc2*-null ELT3 cells reexpressing *Tsc2*; error bars: mean ± SEM in **a–d**, **f** and mean ± SD in **g**, **h**, **k**, **l**; $n = 3$ independent experiments or $n = 6$ in the animal model; *$p < 0.05$, two-tailed, unpaired Student's *t*-tests

(0.25% Triton X-100 in PBS), and stained with the phospho-histone H3-Ser10-Alexa 647 (p-H3; 1:500 in PBS with 3% bovine serum albumin, 9716S; Cell Signaling). After p-H3 staining, cells were washed twice with PBS and suspended in propidium iodide solution. Both cell cycle and p-H3 staining data were acquired on a Gallios Flow Cytometer (Beckman Coulter) at the MD Anderson Cancer Center Flow Cytometry and Cellular Imaging Core Facility and were analyzed by FlowJo v10 software.

**Western blotting**. Total proteins were extracted from cells using urea lysis buffer (8 M urea, 50 mM Tris-Cl pH 7.5, 1% 2-mercaptoethanol, and 1× protease and phosphatase inhibitor cocktails from GenDepot) for total protein. We separated nuclear and non-nuclear fractions using a Dounce homogenizer with Nori buffer (20 mM HEPES pH 7.0, 10 mM KCl, 2 mM $MgCl_2$, 0.5% NP-40, and 1× protease inhibitor cocktail) and urea buffer. Primary antibodies for the western blotting included rabbit anti-mTOR (1:1000, 2983S), rabbit anti-TSC2 (1:1000, 4308S), rabbit anti-p-S6 (1:5000, 2211S), rabbit anti-S6 (1:2500, 2217S), rabbit anti-p-S6K (1:1000, 9234S), rabbit anti-S6K (1:1000, 2708S), rabbit anti-AKT (1:1000, 9272S), rabbit anti-p-AKT (1:1000, 9271S), rabbit anti-cyclin B1 (1:1000, 4138S), rabbit anti-p-H3-S10 (1:1000, 9701S), rabbit anti-γ-H2AX (1:1000, 2577S), rabbit anti-H2AX (1:1000, 2595S), rabbit anti-p-CHK2 (1:1000, 2661S), mouse anti-CHK2 (1:1000, 3440S), rabbit anti-p-CHK1 (1:1000, 2348S), and mouse anti-CHK1 (1:1000, 2360S), all from Cell Signaling Technology; rabbit anti-raptor (1:1000, A300-553A; Bethyl Laboratories); goat anti-rictor (1:1000, sc-50678), mouse anti-PLK1 (1:200, sc-17783), mouse anti-GAPDH (1:1000, sc-32233), mouse anti-TP53 (1:500, sc-99), and mouse anti-MYC (1:5000, sc-40), from Santa Crus Biotechnology; rabbit anti-KDM4B (1:1000 A301-478A and 8639S, from Bethyl Laboratories and Cell Signaling Technology respectively); mouse anti-actin (1:5000, A1978) and mouse anti-α-tubulin (1:5000, T5168), both from Sigma. Secondary antibodies (1:2000) were all purchased from Santa Cruz Biotechnology. Signals were detected with Amersham Enhanced Chemiluminescence Prime Western blotting detection reagent (GE Healthcare Life Sciences). Uncropped blots can be found in Supplementary Figs. 6–18.

**RNA isolation and quantitative reverse transcription PCR**. Complementary DNA was generated from RNA using TRIzol reagent and the SuperScript III kit (Invitrogen). The quantitative PCR reactions were performed using Power SYBR Green PCR Master Mix kit on the ViiA7 Real-Time PCR System (Invitrogen). Quantitative PCR primers were designed to span exon–intron boundaries of respective genes, ensuring that the results were not affected by genomic DNA contamination. The sequences of quantitative PCR primers are listed in Supplementary Table 2.

**Chromatin immunoprecipitation (ChIP)-quantitative PCR assay**. We performed ChIP-quantitative PCR assay following the EZ-ChIP kit instructions (EMD Millipore). U2OS cells were incubated in the growth medium with 1% formaldehyde for 10 min, and we stopped the crosslink reaction with 0.125 M glycine for 5 min at room temperature. We suspended cells in SDS lysis buffer (1% SDS, 10 mM EDTA, 50 mM Tris pH 6.5, and 1× protease inhibitor cocktail) and sheared DNA to around 600 base pairs in length using a 60 Sonic Dismembrator (Fisher Scientific). For each ChIP reaction, we added 900 μl of Dilution Buffer into 100 μl of chromatin and incubated chromatin with protein G agarose beads at 4 °C for 1 h. After centrifugation, 10 μl of the supernatant was removed as input, and the rest of the supernatant was incubated with 1–2 μg antibodies overnight and with protein G agarose beads for 1 h. The antibodies we used for ChIP-quantitative PCR were rabbit anti-KDM4B (8639S; Cell Signaling Technology), ChIP-grade rabbit anti-H3-trimethyl K9 (H3K9me3; ab8898; Abcam), rabbit anti-BMYB (A301-656A; Bethyl Laboratories), and rabbit normal immunoglobulin G (Santa Cruz Biotechnology).

After we washed immunoprecipitation samples with buffers containing different concentrations of salts, we eluted protein–DNA complexes and reversed their crosslinks in all immunoprecipitation samples and in input as well. DNA was purified in 50 μl elution buffer for subsequent quantitative PCR analysis. We added 23 μl of quantitative PCR mix containing 400 nM of primers and Power SYBR Green PCR Master Mix (Life Technologies) to 2 μl of purified DNA for each reaction. The quantitative PCR reactions were performed in the ViiA7 Real-Time PCR System (Invitrogen), and the results of ChIP samples were normalized to input individually in each set of samples. The sequences of CHIP-quantitative PCR primers are listed in Supplementary Table 2.

**Dual-luciferase reporter assay**. U2OS cells were cultured in 60-mm plates to reach 80% confluence, and siRNA oligonucleotides were transfected. One day after siRNA transfection, cells were transfected again with the indicated luciferase-expressing plasmids. For mTOR overexpression, cells were co-transfected with 1 μg mTOR construct, 1 μg firefly luciferase, and 50 ng Renilla luciferase. One day after transfection of luciferase-expressing plasmids, cells were split into six-well plates and incubated for the specified periods of time after 7 Gy of IR. We measured luciferase expression levels driven by the *PLK1* or *CCNB1* promoter (*PLK1*-luciferase or *CCNB1*-luciferase) in 96-well plates using a Dual-Luciferase Reporter

Assay kit (Promega) and a FLUOstar Omega Microplate Reader following the manufacturer's protocol (BMG LABTECH). The sample at each time point was measured in triplicate with normalization to Renilla luciferase activity.

**Single-cell gel electrophoresis (comet assay)**. We treated U2OS cells with 7 Gy of IR, and DNA damage was detected by alkaline comet assay following the comet assay reagent kit instructions (Trevigen). Cells were stained with SYBR Green and then photographed with an Olympus IX81 microscope (Flow Cytometry and Cellular Imaging Core Facility, MD Anderson), and data were analyzed by CometScore v1.6 (TriTek Corp.).

**Cell proliferation assay, colony formation assay, 3D cell culture, and apoptosis assay**. For the MTT (3-(4,5-dimethylthiazol-2-yl)-2,5-diphenyltetrazolium bromide) proliferation assay, we plated 250 or 500 cells per well in 96-well plates one day before drug treatment and incubated the cells with drugs for 4 days. For the colony formation assay, we seeded 500 cells per well in six-well plates one day before drug treatment and cultured cells with drugs for 10 days until visible colonies formed. The medium with drugs were changed every 3 days in order to keep nutrition and drug effects. For 3D cell culture, we seeded 4000 cells per 400 μl of medium in each well of Matrigel pre-coated eight-well chamber slides and cultured the cells for 10 days. We added 0.2 μM MK1775 on day 3 and changed the medium with or without 0.2 μM MK1775 every 3 days. The cells were photographed with an Olympus IX71 microscope (Flow Cytometry and Cellular Imaging Core Facility, MD Anderson) and analyzed by ImageJ. To determine the percentage of apoptotic cells after drug treatment, we used a Gallios Flow Cytometer, following the protocol provided by the manufacturer (Life Technologies), and defined cells positive for annexin V and Alexa Fluor 647 as apoptotic cells. We also directly counted cell numbers 2 days after treatment using a Countess II FL Automated Cell Counter (Thermo Fisher Scientific).

**Immunofluorescence staining**. MEFs were treated with 50 nM MK1775 or/and 50 nM BMN673 for 36 h and fixed with 4% paraformaldehyde (Santa Cruz Biotechnology) for 10 min. We used 3% bovine serum albumin in PBS with Tween-20 as the blocking solution and dilution buffer for all antibodies. Cells were stained with anti-cytochrome c antibody (1:200, BD556432; BD Biosciences) or anti-α-tubulin antibody (1:500, T5168; Sigma-Aldrich) for 2 h and then with Alexa Fluor 594 or 488 anti-mouse secondary antibodies (1:200; Life Technologies) for 2 h, followed by a nuclear counterstain with DAPI (4′,6-diamidino-2-phenylindole)[52]. The whole staining procedure was done at room temperature. Cells were photographed with an Olympus IX81 microscope or an FV1000 laser confocal microscope (Flow Cytometry and Cellular Imaging Core Facility, MD Anderson), and data were analyzed by ImageJ or FV10-ASW v4.2 Viewer (Olympus).

**Animal studies**. All animal works were performed with protocols approved by the MD Anderson Animal Care and Use Committee. $2 \times 10^6$ ELT3-V3-luciferase cells were injected subcutaneously into the posterior flanks of 6-week-old female CB17-SCID mice (Charles River Laboratories). Five weeks after cell injection, mice bearing 100–150 mm³ tumors were randomized into different groups based on the relative equal tumor size right before treatment ($n = 6$)[48]. Mice were treated with MK1775 (in 0.5% methylcellulose, 60 mg kg⁻¹, three times a week) or the vehicle for 3 weeks. Body weights were measured once a week as the parameter of toxicity. Tumor volumes using the formula (length × width²)/2 were measured twice a week. We also used bioluminescence imaging to follow tumor sizes (IVIS 200, MDACC Small Animal Imaging Facility).

**Immunohistochemistry staining**. Formalin-fixed paraffin-embedded ELT3 tumor sections from mice were immersed with citrate buffer (10 mM sodium citrate pH 6.0) in antigen retrieval step. Tumor sections were then treated in peroxidase blocking solution (3% $H_2O_2$), the blocking buffer (5% goat serum and 1% Tween-20 in PBS), anti-cleaved-caspase 3 antibody (1:1000; 9661S; Cell Signaling Technology), SignalStain Boost IHC Detection Reagent (Cell Signaling Technology), and DAB (3,3′-diaminobenzidine) peroxidase substrate (Vector Laboratories). We used hematoxylin as the counterstain. The representative photos were taken with an Olympus BX41 microscope and the data were analyzed by ImageJ.

**Statistical analysis**. Graphs show mean values ± standard deviation (SD) or standard error of the mean (SEM). Statistical significance was determined by two-tailed, unpaired Student's t-tests in GraphPad Prism 6. Results with p-values of <0.05 were considered significant.

# Data availability

All data and codes in this study are included in the article.

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

## Acknowledgements

We thank Associate Professor Dr. Luke K. Wang and Assistant Professor Shan-Chih Hsieh of National Kaohsiung University of Applied Sciences for helpful discussions of matrix generation from the network and MATLAB code. This research was supported by NIH R01 grant CA181663 and Department of Defense grant OC140431 to G.P. G.B.M. thanks the Adelson Medical Research Foundation for a kind gift, and has research support from NCI (1U01CA217842, 1P50CA217685, P50CA098258, 5U54HG008100) Komen Breast Cancer Foundation, Breast Cancer Research Fund, and Ovarian Cancer Research Fund.

## Author contributions

H.-J.H. and G.P. conceived the study. G.B.M., H.-J.H. and G.P. designed experiments. H.-J.H. W.Z., J.S., W.-H.Y. and Y.L. performed experiments. H.-J.H., S.-H.L., J.-Z.W., Y.Z. and H.W. performed mathematical modeling, statistical and bioinformatics analyses. J.Y. provided TSC2 cell lines and discussions. H.-J.H. and G.P. wrote the manuscript. All authors participated in manuscript preparation and approved the final version of the manuscript.

## Additional information

**Competing interests:** G.B.M. is on the S.A.B. of AstraZeneca, has licensed H.R.D. assay to Myriad Genetics, and has research support from Abbie, AstraZeneca, and Tesoro. The remaining authors declare no competing interests.

