## [Peer Review File · Nature Communications]

Reviewers' comments:

Reviewer #1, expert in mTOR and cancer (Remarks to the Author):

The authors used a systems biology approach to identify pathways responsible for G2/M checkpoint activation after DNA damage. The PI3K pathway was one of the strongest regulated. mTORC1-specific inhibition (pharmacological or RNAi) decreased the re-entry into the cell cycle after G2/M checkpoint activation in response to IR. Protein expression of PLK1 and cyclin B1 was affected in mTORC1-depleted cells. Similarly, expression and transcriptional activation of the corresponding genes (PLK1 and CCNB1) was decreased. Importantly, TSC2-null cells exhibited accelerated recovery after G2/M checkpoint activation, increased nuclear KDM4B, and increased expression of PLK1 and cyclin B1. Moreover, they were more sensitive in the combination of a PARP inhibitor (either BMN673 or olaparib) and a Wee1 inhibitor (MK1775). Finally, MK1775 showed moderate inhibition of ELT3 xenograft tumors, compared to vehicle. Overall, this is a strong and novel study, that can greatly influence our way of thinking when treating tumors in TSC and LAM patients, and other neoplasias with mTORC1 hyperactivation in general.

Major concerns.

1. The authors mention that effects of Wee1 inhibitors to the abrogation of G2/M arrest are more pronounced in p53 deficient cancer. Since human TSC tumors are rarely p53 deficient, how Wee1 inhibitors (MK1775) would affect apoptosis in TSC tumors? Of note, the Tsc2 MEFs used in this study are probably genetically deficient for p53 – can the authors confirm that with the source of these cells. If this is the case, the genotype should be corrected throughout the text. p53-dependence also suggests that combination of Wee1 and PARP inhibitors would exert synergy to inhibit TSC tumors, which is the case of their in vitro results. The authors should try monotherapy for MK1775, monotherapy for BMN673, and combination MK1775 / BMN673 in their xenograft animal studies with ELT3 cells.
2. A problem with rapalog therapy is that it does not induce apoptosis, both in in vitro and in in vivo studies. Do MK1775-treated ELT3 xenograft tumors have any evidence of apoptosis? A second caveat with rapalog therapy is that tumors regrow after discontinuation of treatment. Do MK1775-treated tumors re-grow after discontinuation of treatment? My main concern is that MK1775 causes only a modest, although statistically significant, inhibition of ELT3 tumor growth. I would strongly suggest that the authors explore combination of MK1775/BMN673, even MK1775/rapamycin.
3. The authors state that they developed a TSC2-null rapamycin-resistant cell line (ELT3-V3R) by killing 80% of ELT3-V3 cells with 10 nM rapamycin. What was the duration of this treatment? What do they mean by cell "killing", and how was cell "killing" measured? To our experience much higher concentrations of rapamycin (even uM concentrations) do NOT cause apoptosis in a variety of cell lines, including ELT3-V3 cells. Therefore, to my personal opinion, the statement for 80% killing is very inaccurate. Rapamycin merely inhibits cell growth, and causes an S-phase arrest. Upon rapamycin withdrawal cells resume proliferation. What was the duration of the second phase of treatment at the 2.5-20 nM rapamycin? Did they authors use additional methods to characterize ELT3-V3R cells in terms of resistance to rapamycin? Is mTORC1 signaling altered in ELT3-V3R cells, i.e. is rapamycin inducing the same degree of inhibition of mTORC1 kinase activity or dephosphorylation of mTORC1 substrates in ELT3-V3R cells, compared to ELT3-V3 cells? Does rapamycin have an effect in ELT3-V3R cell proliferation or cell number. MTT is probably not the best methods to study viability in cells treated with rapamycin, as this drug has profound effects in mitochondria and metabolism.
4. Presumably ELT3-V3R cells are tumorigenic in SCID mice. Are ELT3-V3R xenograft tumors non-responsive to rapamycin, and how do they compare to ELT3-V3 tumors? Are they sensitive to MK1775, BMN673, or their combination?
5. Mitotic defects, centrosome abnormalities, and increased protein expression of PLK1 in TSC-deficient cells have been previously reported. Also, PLKs seems to interact with multiple

components of the mTOR pathway, including rictor and the TSC1/TSC2 complex. Therefore, it is not surprising that TSC-deficient cells have abnormalities in ploidy and mitotic spindles upon treatment with MK1775 and BMN673, or their combination. Could the authors comment on that? Do the authors believe that the interaction between PLK1 and mTOR pathway components would affect the ability of BMN673 and/or MK1775 to inhibit TSC-null cells?

Minor concerns.

1. In the M&M-Animal Studies section the authors state that mice were treated with MK1775 60mg/kg every two days + BMN673 0.33 mg/kg daily. From the figure legend and main text, this does not seem to be the case. This should be corrected to reflect the data shown in Figure 5, where mice seem to not have been treated only with BMN673.

Reviewer #2, expert in DNA damage repair and G2/M checkpoint (Remarks to the Author):

In this manuscript, authors combined functional proteomics, mathematical modelling and molecular biology approaches and identified mTOR pathway as novel regulator of the checkpoint recovery. They provide evidence that cells are more sensitive to inhibition of mTOR following DNA damage but certain level of mTOR activity is required also for unperturbed cell cycle progression. Further they show that expression of two mitotic inducers CCNB1 and PLK1 is decreased after inhibition of mTOR and propose that this may be due to the increased histone H3K9me3 methylation of its promoters. Finally, authors show that cells lacking TSC2 are more sensitive MK1775 and propose that WEE1 inhibitor could be used as monotherapy in tumours with hyperactivated mTORC1 pathway.

In summary, this study is based on advanced systems biology approaches and addresses an important biological question. Most of the presented data is convincing and supported by various approaches. However, there are also some weak points that should be fully addressed prior publication.

1) Figure 3 shows that KDM4B is localized at CCNB1 and PLK1 promoters and that the level of histone methylation is increased by depletion of mTOR. However, the mechanistic link between mTOR activity and KDM4B has not been fully resolved. If authors are right that KDM4B is the major factor acting downstream of mTOR and determining the rate of checkpoint recovery, they could demonstrate that overexpression of KDM4B rescues the recovery defect observed after depletion or inhibition of mTOR.

2) Although the observation of an increased sensitivity of TSC2 deficient cells to WEE1 inhibitor is interesting, it cannot be easily explained by accelerated recovery due to the active mTOR. Inhibition of WEE1 on its own allows full activation of CDK1, impairs checkpoint activation and leads to cell death by mitotic catastrophe. Ionizing radiation, rather than WEE1 inhibitor, may be used to study the effect of the checkpoint override caused by the loss of TSC2 on the viability of the cells.

3) Figure 5 shows that MK1775 treatment to some extent slows down the growth of ELT3-V3 cells in a xenograft model. However, this study did not address whether this level of growth inhibition was sufficient to eradicate the tumour and to prolong survival. Also it is unclear which clinically relevant tumours might be sensitive to this inhibitor. Therefore, all statements about the therapeutical use of WEE1 inhibitor in this context should be removed from the manuscript.

Reviewer #3, expert in protein arrays and bioinformatics (Remarks to the Author):

The manuscript, entitled "Systems biology approach reveals that mTORC1 regulates G2/M DNA damage checkpoint recovery, creating a therapeutic vulnerability in mTOR-hyperactivated tumors" by Hsieh et al., describes the use of reverse protein arrays as an entry point to identify important downstream signaling components that govern the G2/M checkpoint recovery in cells. The authors next employed an integrated approach and identified mTORC1 as a critical determinant for recovery from G2/M checkpoint. With more detailed cell-based studies, other downstream components, such as CCNB1 and PLK1, were further characterized for their roles during the recovery. Because mTOR is known to play an important role in tumorigenesis, the authors argued that their discovery might help develop new drugs targeting mTOR as a new therapeutic target in cancers.

In general, this is a solid study, especially in later parts of the manuscript. My major concern is the part of the description of the use of the reverse phase protein array technology. I cannot find a clear description of this part in either the main text or Methods section. I had to dig into the Excel spreadsheet to get an idea how the assays were designed and executed. Apparently, the authors employed antibodies targeting 172 proteins to examine changes in protein expression level on the reverse phase protein arrays. However, it is totally not clear how these 172 protein targets were selected. Are they somehow enriched in relevant GO terms? Or, are they known to have elevated mRNA levels during the process of G2/M checkpoint recovery? More importantly, how did the authors select these 199 antibodies? I only saw 35 of these antibodies were generated from mice, which I would assume that they are monoclonal antibodies. For the rest, I would assume the majority of them were polyclonal antibodies. If this is the case, how did the authors gauge the quality of these antibodies? This is an important issue because it is well known that the majority of polyclonal antibodies are not very specific. I also found perplexing that most, if not all, of their immunoblot analyses only showed a narrow area of the blots without any labeling of expected MWs of the protein bands. Therefore, I suggest that the authors provide convincing evidence for the specificity of the antibodies used in this study and show a larger area of their IB blots as supplemental figures with clear labeling of the MWs.

MS# NCOMMS-17-15399-T

Systems biology approach reveals that mTORC1 regulates G2/M DNA damage checkpoint recovery, creating a therapeutic vulnerability in mTOR-hyperactivated tumors

Dear Editors and Reviewers,

Thank you for all your time and work on our manuscript.

We really appreciate support from the reviewers on the significance and the potential impact of our findings and their constructive suggestions to further improve our study. Here we provide the detailed point-to-point response to reviewers' comments with new experimental data. We believe that our work will not only help us understand the new function of mTOR in regulating DNA damage checkpoint recovery, but also help us further develop study to demonstrate whether tumors with hyper-mTOR activity may benefit from WEE1 inhibitors.

To facilitate the review of our rebuttal letter and manuscript, we present all the new data as rebuttal letter figures in this letter, with referrals to corresponding figures and text in our revised manuscript. We have also marked all the changes in our revised manuscript by colored text.

Reviewer #1: expert in mTOR and cancer

“The authors used a systems biology approach to identify pathways responsible for G2/M checkpoint activation after DNA damage. The PI3K pathway was one of the strongest regulated. mTORC1-specific inhibition (pharmacological or RNAi) decreased the re-entry into the cell cycle after G2/M checkpoint activation in response to IR. Protein expression of PLK1 and cyclin B1 was affected in mTORC1-depleted cells. Similarly, expression and transcriptional activation of the corresponding genes (PLK1 and CCNB1) was decreased. Importantly, TSC2-null cells exhibited accelerated recovery after G2/M checkpoint activation, increased nuclear KDM4B, and increased expression of PLK1 and cyclin B1. Moreover, they were more sensitive in the combination of a PARP inhibitor (either BMN673 or olaparib) and a Wee1 inhibitor (MK1775). Finally, MK1775 showed moderate inhibition of ELT3 xenograft tumors, compared to vehicle.”

“Overall, this is a strong and novel study, that can greatly influence our way of thinking when treating tumors in TSC and LAM patients, and other neoplasia with mTORC1 hyperactivation in general.”

We thank Reviewer #1 for all his/her positive comments and support on our study.

Major concerns.

1. *“The authors mention that effects of Wee1 inhibitors to the abrogation of G2/M arrest are more pronounced in p53 deficient cancer. Since human TSC tumors are rarely p53 deficient, how Wee1 inhibitors (MK1775) would affect apoptosis in TSC tumors? Of note, the Tsc2 MEFs used in this study are probably genetically deficient for p53 – can the authors confirm that with the source of these cells. If this is the case, the genotype should be corrected throughout the text.*

p53-dependence also suggests that combination of Wee1 and PARP inhibitors would exert synergy to inhibit TSC tumors, which is the case of their in vitro results.”

In our original manuscript (*Discussion* section), based on previous publications (Vakifahmetoglu, H., et al. *Cell Death Differ*, 2008; Vitale, I., et al. *Nat Rev Mol Cell Biol*, 2011), we mentioned that “the effect of WEE1 inhibition is more pronounced in p53-deficient cancers, which have a G1 checkpoint defect and strongly depend on the G2 checkpoint to prevent cell cycle progression”. This is one mechanism that may lead to an increased sensitivity to WEE1 inhibitors. In our study, we discovered a new mechanism that can potentially sensitize cells to WEE1 inhibitors. In TSC2-deficient cells, constitutive high mTORC1 activity leads to accelerated G2/M transition, particularly in the presence of DNA damage induced by ionizing radiation (IR) or PARPi treatment. The WEE1 inhibitor further abolishes the G2/M checkpoint and leads to increased mitotic catastrophe in TSC2-deficient cells and shows synergy in combination with PARPi and IR [**Rebuttal Figs. 1-3; 8 and 10 (revised Supplementary Figure 4c-4g, Figure 5f, and Supplementary Figure 5d-5f)**]. Thus, our study provides a mechanistic rationale for using G2/M checkpoint inhibitors such as WEE1 inhibitors to target cells with hyper-mTORC1 activity such as TSC2-deficient cells. Based on this new mechanism, the therapeutic effect of WEE1 inhibition on TSC2-deficient cells is not dependent on p53 status. Notably, a variety of signaling pathways have been shown to mediate DNA damage-induced apoptosis in a p53-independent manner (McNamee L. et al. *Genetics* 2009; Wichmann A., et al. *Proc Natl Acad Sci USA*, 2006). We conducted new experiments to further support this observation.

As suggested by the reviewer, we first confirmed that *Tsc2*-knockout MEFs (TSC2^{-/-}) are indeed genetically deficient of p53, which were the only p53-deficient cells used in our study [**Rebuttal Fig. 1a (revised Supplementary Fig. 4c)**]. We then examined apoptosis in these MEFs and found that TSC2-deficient cells (TSC2^{-/-} p53^{-/-}) were still more sensitive to WEE1i (MK1775), PARPi (BMN673) and the combination treatment [**Rebuttal Fig. 1b (revised Supplementary Fig. 4d)**]. Furthermore, we examined apoptosis in p53-proficient ELT3 cells. ELT3-V3 (TSC2^{-/-} cells) and ELT3-V3R (TSC2^{-/-} cells resistant to rapamycin treatment) were more sensitive to the treatment compared to ELT3-T3 (TSC2-reconstituted cells) [**Rebuttal Fig. 1c (revised Supplementary Fig. 4e and 4g)**]. Moreover, we examined expression of a key apoptosis marker, cleaved caspase-3, in ELT3-V3 xenograft tumors from new *in vivo* experiments that we conducted during the revision (please also see our response to *Reviewer #1: Comment 2*) [**Rebuttal Fig. 2 (revised Fig. 5f)**]. Consistent with *in vitro* data, MK1775, BMN673 and the combination treatment significantly induced apoptosis in TSC2-null tumors. As expected, rapamycin treatment did not induce apoptosis. Interestingly combination of rapamycin and MK1775 induced apoptosis. Collectively these data support our finding that TSC2-deficient cells are more sensitive to WEE1 inhibition, which is independent of p53 status. In the revised manuscript, we presented these new data to strengthen our observation in the *Result* section and removed the original statement from the *Discussion* section.

Rebuttal Fig. 1. TSC2 deficiency sensitizes cells to WEE1i, PARPi and the combination treatment in a p53-independent manner. (a) Western blots show p53 and TSC2 expression. Cells were exposed to IR (7 Gy). (b) Apoptosis assay shows an increased sensitivity of TSC2-deficient cells to the indicated treatments in a p53-depleted genetic context. (c) Apoptosis assay shows an increased sensitivity of TSC2-deficient cells with proficient p53 to the indicated treatments.

Rebuttal Fig. 2. Analysis of apoptosis in TSC2-deficient xenograft tumors. Tissue sections were stained with cleaved caspase 3 and hematoxylin and the graph represents the percentage of cleaved caspase 3 positive cells in each group.

2. “The authors should try monotherapy for MK1775, monotherapy for BMN673, and combination MK1775 / BMN673 in their xenograft animal studies with ELT3 cells.”

“A second caveat with rapalog therapy is that tumors regrow after discontinuation of treatment. Do MK1775-treated tumors re-grow after discontinuation of treatment? My main concern is that MK1775 causes only a modest, although statistically significant, inhibition of ELT3 tumor growth. I would strongly suggest that the authors explore combination of MK1775/BMN673, even MK1775/rapamycin.”

As shown in our manuscript **Fig. 4** and **Supplementary Fig. 4**, the combination treatment using MK1775 and BMN673 exhibited a better therapeutic effect than monotherapy. As suggested by the reviewer, we first tested the therapeutic effects of combination using MK1775 and rapamycin in two *in vitro* cell models: MEFs and ELT3 rat cells. To exclude potential effects of rapamycin on MTT assay due to metabolic changes, we directly counted the cell number by trypan blue staining and found that TSC2-deficient cells (either TSC2 knockout MEFs or TSC2-null ELT3-V3 cells) were more sensitive to the combination treatment [**Rebuttal Fig. 3a and 3b (revised**

Supplementary Fig. 5d and 5e)]. We further performed colony formation assay in ELT3 cells. TSC2-deficient cells were more sensitive to MK1775 alone, rapamycin alone, and MK1775 combined with rapamycin treatment [Rebuttal Fig. 3c and 3d (revised Supplementary Fig. 5f)].

We then conducted *in vivo* experiments to test the monotherapy and combination therapy using MK1775, BMN673 and rapamycin (Rebuttal Fig. 4). Our data showed that tumors formed by this TSC2-deficient ELT3 cell line were very sensitive to rapamycin treatment (Rebuttal Fig. 4). In this rapamycin-sensitive xenograft tumor model, consistent with our previous result (Fig. 5a-e), MK1775 and BMN673 treatment reduced tumor growth (Rebuttal Fig. 4). However, we did not observe synergy in the combination treatment using MK1775 and BMN673 or using rapamycin and MK1775. Notably, there was no apparent toxicity in either monotherapy or combination therapy as no significant changes in mice body weight were observed (Rebuttal Fig. 4d).

As suggested by the reviewer, in addition to examining tumor growth after 4 weeks of treatment, we monitored the regrowth of tumors when all treatments were stopped at 28 days (**Rebuttal Fig. 5a**). Regrowth of tumors treated with MK1775, BMN673 and the combination was slower than control tumors. As discussed above, we did not observe a synergy in combination treatment in tumor regrowth analysis. As shown in **Rebuttal Fig. 5b**, this tumor model was very sensitive to rapamycin treatment, which significantly extended mice survival after 28 days of treatment compared to the control mice (died around 40 days). We did observe that MK1775 and rapamycin combination further extended survival of some mice, although these tumors already were very sensitivity to rapamycin monotherapy. Due to the large variation and limited number of mice tested (n=6 per group), the result did not achieve statistical significance. Nevertheless, these data suggest a new direction for our future research to explore whether MK1775 monotherapy or in combination with PARPi or rapamycin might slow regrowth of tumors after drug withdraw in addition to inhibiting tumor growth.

There are several limitations in our *in vivo* experiments that may explain the lack of synergy in the combination treatment.

First, in the *in vivo* study, we tested only one dosage for each drug in the combination treatment, which is normally used for monotherapy. It could be possible that the combination treatment may have therapeutic benefits when both drugs were used in a lower concentration. The optimal combination dosage needs to be further investigated.

Second, it is possible that the sequence of drug treatment may affect the combination efficacy. In our current experiment, we treated mice with the combination therapy by using WEE1i and PARPi at the same time and by using rapamycin and WEE1i at the same time. PARPi induces DNA damage, while WEE1i removes the G2/M checkpoint and drives cell cycle progression without leaving sufficient time for cells to repair damaged DNA. Thus, to treat mice first with

PARPi and then followed by WEE1i may represent a better strategy to achieve therapeutic benefits. In addition, to treat mice with rapamycin first, then withdraw rapamycin and treat with MK1775 may also exhibit a better therapeutic effect than administration of two drugs at the same time. When two drugs are given at the same time, rapamycin arrests cells in G1 phase, which may actually block the therapeutic effect of MK1775, which mainly targets G2/M cell cycle transition. If rapamycin is given first and then withdrawn, cells will go to S and M phases. At this time point, MK1775 may cause uncontrolled/unscheduled mitosis entry, which induces apoptosis more efficiently. Thus, due to the unique effect of rapamycin and MK1775 on cell cycle progression, it would be important to determine whether the combination should be given together or in sequence and if the sequential treatment might be more beneficial, which drug should be given first.

Third, this tumor model is rapamycin sensitive. It could be a challenge to examine a synergistic effect when rapamycin alone strongly suppressed tumor growth. It raises another question that different mTOR hyperactivation tumor models might be used in testing the combination for example rapamycin-sensitive vs rapamycin-resistant tumors. It is possible that WEE1i and PARPi combination or WEE1i and rapamycin combination may have more therapeutic benefits in treating rapamycin-insensitive tumors.

Forth, as discussed earlier, the combination of MK1775 and BMN673 or rapamycin may exert a synergistic effect on delaying/blocking tumor regrowth after drug withdraw. Thus, tumor regrowth is an important aspect to determine synergy in addition to measuring tumor growth during the treatment.

Our current study focused on using multiple-disciplinary approaches including functional proteomics, systems biology, mathematical modeling and molecular biology to identify molecular signaling required for DNA damage G2/M checkpoint recovery. We discovered a new mechanism that mTORC1 signaling regulates G2/M checkpoint recovery. The research questions/directions discussed above are related to developing translational and preclinical studies for tumors with hyper-mTOR activity. It is our logic next step to address these important questions, which however are beyond the scope of our current manuscript. We want to thank the reviewer for the constructive comments. Based on his/her comments, the new *in vivo* experiments we conducted during the revision opened new directions for our future study toward translational research. As we are aware of limitations of our current *in vivo* experiments and also suggested by the reviewer #2, we have removed the conclusive statement about the therapeutic effects in our manuscript. It is our hope that after we publish this paper, we will be able to obtain funding to support systematic *in vivo* experiments as discussed above.

“A problem with rapalog therapy is that it does not induce apoptosis, both in in vitro and in in vivo studies. Do MK1775-treated ELT3 xenograft tumors have any evidence of apoptosis?”

As shown in **Rebuttal Fig. 2 (revised Fig. 5f)** and also *Response to Comment 1*, we treated mice with ELT3 tumors with MK1775, BMN673, rapamycin or combination for four weeks. Tumors were analyzed by immunohistochemistry (IHC) staining of cleaved-caspase 3. The expression of

cleaved caspase 3 was significantly higher in tumors treated with MK1775 alone, BMN673 alone, BMN673 combined with MK1775, and rapamycin combined with MK1775 compared to the expression in tumors treated with the vehicle control. Consistent with the reviewer’s comment and previous reports, rapamycin did not induce apoptosis in tumors. These data indicated apoptosis in MK1775-treated ELT3 xenograft tumors.

3. “The authors state that they developed a TSC2-null rapamycin-resistant cell line (ELT3-V3R) by killing 80% of ELT3-V3 cells with 10 nM rapamycin. What was the duration of this treatment? What do they mean by cell “killing”, and how was cell “killing” measured? To our experience much higher concentrations of rapamycin (even uM concentrations) do NOT cause apoptosis in a variety of cell lines, including ELT3-V3 cells. Therefore, to my personal opinion, the statement for 80% killing is very inaccurate. Rapamycin merely inhibits cell growth, and causes an S-phase arrest. Upon rapamycin withdrawal cells resume proliferation. What was the duration of the second phase of treatment at the 2.5-20 nM rapamycin?”

We thank the reviewer for this comment. Rapamycin indeed exhibited ‘growth inhibition’ rather than ‘killing’. The change has been made in the manuscript to reflect a more accurate description. To develop TSC2-null rapamycin-resistant cell line (ELT3-V3R), we cultured TSC2-null ELT3-V3 cells in a very low density (500 cells/well in the 6-well plate) with 10 nM rapamycin for one week. We then sub-cultured the remaining cells as a pool and treated cells with rapamycin from low concentration to high concentration (increased concentration every passage from 2.5 nM, 5 nM, 10 nM to 20 nM, approximately every three or four days). The ELT3-V3R cells were then maintained in regular culture medium with 20 nM rapamycin.

“Did they authors use additional methods to characterize ELT3-V3R cells in terms of resistance to rapamycin? Is mTORC1 signaling altered in ELT3-V3R cells, i.e. is rapamycin inducing the same degree of inhibition of mTORC1 kinase activity or dephosphorylation of mTORC1 substrates in ELT3-V3R cells, compared to ELT3-V3 cells? Does rapamycin have an effect in ELT3-V3R cell proliferation or cell number. MTT is probably not the best methods to study viability in cells treated with rapamycin, as this drug has profound effects in mitochondria and

metabolism.”

To further characterize rapamycin-resistant ELT3-V3R cells, we conducted reverse phase protein array (RPPA) to analyze altered molecular signaling in rapamycin-resistant ELT3-V3R cells (*Methods* section). We treated cells with indicated concentrations of rapamycin for 16 hours and subjected cell lysis for RPPA analysis. We generated the heat map with hierarchical clustering focusing on antibodies related to the mTOR signaling pathway [**Rebuttal Fig. 6a (revised Supplementary Fig. 5h)**]. Expression of mTOR-related signaling proteins exhibited no remarkable changes in different ELT3 cells (V3: TSC2 null; T3: TSC2 reconstituted; V3R: TSC2 null-rapamycin resistance). After rapamycin treatment, the expression pattern in ELT3-V3R cells was in general very similar to the pattern in its parental ELT3-V3 cells. Notably, rapamycin reduced phosphorylation of S6 more significantly in TSC2 null cells (V3 and V3R) compared to TSC2-reconstituted cells (T3). Interestingly, phosphorylation of AKT (serine 473) was significantly higher in ELT3-V3R cells after rapamycin treatment compared to parental ELT3-V3 cells, particularly at a high concentration (20 nM). These results were further confirmed by western blot analysis [**Rebuttal Fig. 6b (revised Supplementary Fig. 5g)**] and were consistent with previous publications that p-AKT and activation of the TORC2 complex play an important role in rapamycin resistance (*Gulhati P. et al. Clinical Cancer Research 2009; Mi W. et al., Oncotarget 2015; Yoon SO et al., Current Biology 2013*). Collectively, our data showed that rapamycin induces a very similar pattern of mTOR-related signaling in ELT3-V3R cells compared to ELT3-V3 cells. ELT3-V3R cells showing resistance to rapamycin, potentially due to activation of p-AKT, could still remain sensitive to WEE1i, which targets TSC2-null cells

by a different mechanism compared to rapamycin.

As suggested by the reviewer, MTT may not be the best method to study viability in cells treated with rapamycin, as this drug has a profound effect in mitochondria and metabolism. Therefore, we also examined cell proliferation and growth by using colony formation assay to test the sensitivity to rapamycin. ELT3-V3R cells exhibited a reduced sensitivity to rapamycin compared to its parental TSC2-null (ELT3-V3) cells [Rebuttal Fig. 7a (revised Supplementary Fig. 5b)]. We released ELT3-V3R cells from 20 nM rapamycin for three passages (ELT3-V3R were cultured without rapamycin for 2 weeks). These cells showed a very similar sensitivity to rapamycin compared to ELT3-V3R cells that were constitutively cultured in 20 nM rapamycin [Rebuttal Fig. 7b (revised Supplementary Fig. 5c)]. These data indicated that resistance to rapamycin in ELT3-V3R cells can be retained even in the absence of maintenance rapamycin in cell culture. The resistant trait of this cell line was relatively stable.

4. “Presumably ELT3-V3R cells are tumorigenic in SCID mice. Are ELT3-V3R xenograft tumors non-responsive to rapamycin, and how do they compare to ELT3-V3 tumors? Are they sensitive to MK1775, BMN673, or their combination?”

We first tested drug sensitivity *in vitro* and found that both parental TSC2-null ELT3-V3 cells and rapamycin-resistant ELT3-V3 cells were sensitive to MK1775, BMN673, and the combination compared to TSC2-rescued ELT3-T3 cells by colony formation assay [Rebuttal Fig. 8 (revised Supplementary Fig. 4f)] and apoptotic assay [Rebuttal Fig. 1c (revised Supplementary Fig. 4e)].

We then conducted xenograft assay using ELT3-V3R resistant cells. We requested an extension of our revision from 3 months to 5 months in order to develop rapamycin-

resistant ELT3-V3R tumors. However, these rapamycin cells exhibited a very slow growth rate (possibly developed senescence phenotype after several passages), that limited the usage of these cells for *in vivo* experiment. We were unable to expand sufficient cells for *in vivo* injection as conducted in Rebuttal Fig. 4 and 5. Thus, we injected a much smaller amount of ELT3-V3R

cells, which did not develop tumors suitable for completing drug treatment experiment. We propose to take an alternative approach to develop rapamycin-resistant lines from mice bearing residual ELT3-V3 tumors after rapamycin treatment by using a series of *in vivo* transplantation methods. As discussed in our response to *Comment 2*, we sincerely hope that we can publish our current mechanistic findings and obtain essential funding to support systematic *in vivo* studies in near future.

5. *“Mitotic defects, centrosome abnormalities, and increased protein expression of PLK1 in TSC-deficient cells have been previously reported. Also, PLKs seems to interact with multiple components of the mTOR pathway, including rictor and the TSC1/TSC2 complex. Therefore, it is not surprising that TSC-deficient cells have abnormalities in ploidy and mitotic spindles upon treatment with MK1775 and BMN673, or their combination. Could the authors comment on that? Do the authors believe that the interaction between PLK1 and mTOR pathway components would affect the ability of BMN673 and/or MK1775 to inhibit TSC-null cells?”*

We thank the reviewer for this comment. As the reviewer pointed out that TSC2-deficient cells showed increased PLK1 expression and additionally PLK1 interacts with multiple components of the mTOR pathway (Astrinidis A. *et al.*, *Human Molecular Genetics*, 2006; Valianou M. *et al.*, *Cell Cycle* 2014; Ruf S. *et al.*, *Autophagy* 2017; Li Z. *et al.*, *Cancer Research* 2018). These studies were primarily conducted in cells without exogenous DNA damage and were associated with the coordination/cooperation of mTOR signaling and PLK1 signaling in regulating unperturbed cell proliferation/cell cycle progression. In contrast, our study uncovers the role of the mTOR signaling in regulating recovery of cell cycle from exogenous DNA damage-induced G2/M arrest. A severe defect in G2/M recovery was observed in mTOR-deficient cells after exogenous DNA damage. In contrast, only a minor change in G2/M transition was observed in unperturbed cell proliferation. We further validated our findings in TSC2-null cells. As expected, TSC2-null cells also showed a dramatic increased G2/M recovery in response to exogenous DNA damage compared. In contrast, no significant changes of G2/M transition were observed in TSC2-null cells in unperturbed cell cycle (absence of DNA damage). Furthermore, we identified a new role of KDM4B in regulating DNA damage-induced PLK1 and cyclin B1 expression during G2/M checkpoint recovery, which is distinct from PLK1-mTOR signaling in regulating unperturbed cell cycle.

We completely agree with the reviewer that it is possible that the interaction between PLK1 and components of the mTOR pathway would affect the ability of BMN673 and/or MK1775 to inhibit TSC-null cells. It remains to be determined whether DNA damage regulates the interaction between PLK1 and components of the mTOR pathway. DNA damage can induce a variety of post-translational modifications such as phosphorylation. It is very likely that these DNA damage-induced protein modifications may alter PLK1-mTOR interactions, which may also contribute to response of TSC-null cells to MK1775 and/or BMN673. We have cited the original publications and the discussion to Discussion section.

Minor concerns.

1. *“In the M&M-Animal Studies section the authors state that mice were treated with MK1775*

60mg/kg every two days + BMN673 0.33 mg/kg daily. From the figure legend and main text, this does not seem to be the case. This should be corrected to reflect the data shown in Figure 5, where mice seem to not have been treated only with BMN673.”

We thank the reviewer for this comment. We apologize for the incorrect description in Method section. The correct information has been presented in the Method section: ‘Mice were treated with MK1775 (in 0.5% methylcellulose, 60mg/kg, three times a week)’.

In Rebuttal Fig. 4 and Fig. 5, mice were treated for combination therapy: Mice were treated with MK1775 (in 0.5% methylcellulose, 60mg/kg, three times a week); BMN 673 (in 5% dimethylacetamide, 5% Solutol and 85% PBS, 0.33mg/kg, five times a week); rapamycin (in 6% ethanol, 0.25% PEG-400, and 0.25% Tween-80, 3mg/kg, three times a week); MK1775 combined with BMN673; or MK1775 combined with rapamycin for four weeks (28 days).

Reviewer #2, expert in DNA damage repair and G2/M checkpoint (Remarks to the Author):

“In this manuscript, authors combined functional proteomics, mathematical modelling and molecular biology approaches and identified mTOR pathway as novel regulator of the checkpoint recovery. They provide evidence that cells are more sensitive to inhibition of mTOR following DNA damage but certain level of mTOR activity is required also for unperturbed cell cycle progression. Further they show that expression of two mitotic inducers CCNB1 and PLK1 is decreased after inhibition of mTOR and propose that this may be due to the increased histone H3K9me3 methylation of its promoters. Finally, authors show that cells lacking TSC2 are more sensitive MK1775 and propose that WEE1 inhibitor could be used as monotherapy in tumours with hyperactivated mTORC1 pathway.

In summary, this study is based on advanced systems biology approaches and addresses an important biological question. Most of the presented data is convincing and supported by various approaches. However, there are also some weak points that should be fully addressed prior publication.”

We thank the reviewer for his/her positive comments and support on our study.

1. “Figure 3 shows that KDM4B is localized at CCNB1 and PLK1 promoters and that the level of histone methylation is increased by depletion of mTOR. However, the mechanistic link between mTOR activity and KDM4B has not been fully resolved. If authors are right that KDM4B is the major factor acting downstream of mTOR and determining the rate of checkpoint recovery, they could demonstrate that overexpression of KDM4B rescues the recovery defect observed after depletion or inhibition of mTOR.”

Based on the reviewer's comment, we overexpressed different level of KDM4B in mTOR-depleted U2OS cells by transfecting different amount of FLAG-KDM4B plasmid. As shown in **Rebuttal Fig. 9 (revised Supplementary Fig. 3f and 3g)**, after IR and paclitaxel treatment,

KDM4B overexpression could rescue cyclin B1 expression after DNA damage in mTOR-knockdown cells. These data suggest that KDM4B plays an important role, downstream of mTOR, in determining checkpoint recovery.

2. *“Although the observation of an increased sensitivity of TSC2 deficient cells to WEE1 inhibitor is interesting, it cannot be easily explained by accelerated recovery due to the active mTOR. Inhibition of WEE1 on its own allows full activation of CDK1, impairs checkpoint activation and leads to cell death by mitotic catastrophe. Ionizing radiation, rather than WEE1 inhibitor, may be used to study the effect of the checkpoint override caused by the loss of TSC2 on the viability of the cells.”*

We thank the reviewer for this comment and also the constructive suggestion. Our data (**Figure 4a-d**) have shown that TSC2-depleted cells exhibited an accelerated G2/M recovery after IR. Thus, we conducted experiments with ionizing radiation (IR) rather than WEE1 inhibition to study the effect of checkpoint override caused by the loss of TSC2 on the viability of the cells. MEFs were treated with different dosages of IR. As expected, we found TSC2-depleted cells had higher percentage of apoptotic cells compared to the percentage in wild-type cells (**Rebuttal Fig. 10a**). It is well known that MEFs are relative resistant to IR-induced apoptosis. Compared to IR at 7.5 Gy, IR at 15 Gy induced remarkable mitotic catastrophe including more multipolar mitosis and multinucleated cells in TSC2-depleted MEFs (**Rebuttal Fig. 10b**). These data indicate that the G2/M checkpoint override due to TSC2 deficiency reduces the viability of cells to IR.

3. “Figure 5 shows that MK1775 treatment to some extent slows down the growth of ELT3-V3 cells in a xenograft model. However, this study did not address whether this level of growth inhibition was sufficient to eradicate the tumour and to prolong survival. Also it is unclear which clinically relevant tumours might be sensitive to this inhibitor. Therefore, all statements about the therapeutic use of WEE1 inhibitor in this context should be removed from the manuscript.”

We completely agree with the reviewer. We have removed the conclusive statements about the therapeutic use of WEE1 inhibitor as suggested.

Reviewer #3, expert in protein arrays and bioinformatics (Remarks to the Author):

“The manuscript, entitled “Systems biology approach reveals that mTORC1 regulates G2/M DNA damage checkpoint recovery, creating a therapeutic vulnerability in mTOR-hyperactivated tumors” by Hsieh et al., describes the use of reverse protein arrays as an entry point to identify important downstream signaling components that govern the G2/M checkpoint recovery in cells. The authors next employed an integrated approach and identified mTORC1 as a critical

determinant for recovery from G2/M checkpoint. With more detailed cell-based studies, other downstream components, such as CCNB1 and PLK1, were further characterized for their roles during the recovery. Because mTOR is known to play an important role in tumorigenesis, the authors argued that their discovery might help develop new drugs targeting mTOR as a new therapeutic target in cancers.

In general, this is a solid study, especially in later parts of the manuscript. My major concern is the part of the description of the use of the reverse phase protein array technology. I cannot find a clear description of this part in either the main text or Methods section. I had to dig into the Excel spreadsheet to get an idea how the assays were designed and executed. Apparently, the authors employed antibodies targeting 172 proteins to examine changes in protein expression level on the reverse phase protein arrays. However, it is totally not clear how these 172 protein targets were selected. Are they somehow enriched in relevant GO terms? Or, are they known to have elevated mRNA levels during the process of G2/M checkpoint recovery? More importantly, how did the authors select these 199 antibodies? I only saw 35 of these antibodies were generated from mice, which I would assume that they are monoclonal antibodies. For the rest, I would assume the majority of them were polyclonal antibodies. If this is the case, how did the authors gauge the quality of these antibodies? This is an important issue because it is well known that the majority of polyclonal antibodies are not very specific. I also found perplexing that most, if not all, of their immunoblot analyses only showed a narrow area of the blots without any labeling of expected MWs of the protein bands. Therefore, I suggest that the authors provide convincing evidence for the specificity of the antibodies used in this study and show a larger area of their IB blots as supplemental figures with clear labeling of the MWs.”

Reverse Phase Protein Array (RPPA) is a standard proteomic technology provided by MD Anderson Cancer Center Functional Proteomics Core Facility (<https://www.mdanderson.org/research/research-resources/core-facilities/functional-proteomics-rppa-core.html>). The antibodies used in our study are the standard set of antibodies developed and used by the core facility, which covers major signaling transduction pathways involved in cancer biology. We chose this technology because it is a high-throughput, quantitative and cost-efficient method and allows us to analyze molecular signaling involved in DNA damage checkpoint recovery.

As pointed out by the reviewer, the validation of antibodies for RPPA analysis is one of the most important steps for the success of RPPA technology. The antibodies used in our study have been thoroughly validated for their specificity by the core facility. This method has been used/described in several recent publications from our collaborator Dr. Gordon Mills' group (*Sun C. et al., Cancer Cell 2018; Zhang, Y. et al., Cancer Cell 2017; Li J. et al, Cancer Cell 2017; Zhao W., et al., Oncoscience 2017*). Briefly we summarize the validation process here. First, the levels of protein expression derived from RPPA correlate with the density of the appropriately-sized single band on immunoblots (a Pearson correlation coefficient $R \geq 0.7$). In the validation process, a variety of methods are used to alter protein expression especially for phosphorylated proteins including peptides, phosphopeptides, inhibitors, growth factors or siRNA. Second,

samples on the same and different slides (intra- and inter-slide) shows that the functional proteomic “fingerprint” is reproducible from RPPA. Antibodies with coefficients of variation that are not consistently <15% will not be used for RPPA. Third, the concordance between protein expression levels and mRNA expression levels provides additional validation. Antibodies with a poor protein-mRNA correlation will not be used.

We have scanned and added all uncropped western blots as below (**revised Supplementary Figure 6-18**) Blots were labeled with predicted molecular weights based on the protein marker (GenDEPOT). We also marked the protein bands used in the manuscript in black.

Supplementary Figure 6: 7 8

Figure 2f

C TOR KD
 0 8 16 24 0 8 16 24 hrs IR 7 Gy

Supplementary Figure 7:

12. hel-Ne-Xuh:
 1) 1/6 sample

$\left\{ \begin{array}{l} \text{Si-NT} \\ \text{Si-Rap} \\ \text{Si-Ric} \end{array} \right. + 77 \quad \frac{0}{8} \quad \frac{16}{24} \quad \frac{77}{16/24/77}$

Figure 2k

Supplementary Figure 8:

Supplementary Figure 2c

1 2 3
mock siRNA
Figure 2b Ctrl mTOR

Supplementary Figure 9:

Si-mTOR - 1# / 2# (300 pmol / 100m plate)
 will shared slow Recovery phenotype

1# seemed very similar to Si-mTOR-pool, but 2# have significantly higher end-level of mTOR

Supplementary Figure 2c

Supplementary Figure 2e

1 2
 C rapa
 20nM

Supplementary Figure 10: 6 7 8 HCT 116

ctrl mTOR KD #193
 0 3 6 7 0 3 6 9 IR (hy)

Supplementary Figure 2d

1 2 3 4 5 6 7 8 9 10
 C ZOR KD
 0 1 4 8 16 0 1 4 8 16 hrs IR 7 Gy

Supplementary Figure 3b

Supplementary Figure 11:

Supplementary Figure 3d

12/30. HCT116.
 1 2 3 4 5 6
 0 3 12 0 3 12 hrs
 C FDM48
 KD Taxol

Supplementary Figure 3e

1 2 3 4
 C MK2206
 0.1μM 28hrs.
 - + - + 2R 24 hrs + Taxol

Supplementary Figure 3h

Supplementary Figure 12:

mTOR KD + KDM4B OE.

1 2 3 4 5 6 7 8
 (1,2) ctrl (3,4) mTOR
 3x flag 3x flag KDM4B
 1x 5x

- + - + - + - + IR 9Gy 24hrs
 + Taxol.

Supplementary Figure 3f

1 2 3 4
 C rapa c rapa 36hrs 20nM
 MG132 20uM 2hrs

Supplementary Figure 3i

Supplementary Figure 13:

1/28. 1 2 3 4 5 6 7 8 9 10 11 12

C SiKDM4B SiMTOR IR 1 Gy + Taxol.

0 8 16 24 0 8 16 24 0 8 16 24

Figure 3e

Supplementary Figure 14:

Figure 3f

Supplementary Figure 15:

Figure 4b

Supplementary Figure 16:

1 2 3 4 5 6 7 8 9 10
 +/+ -/- MEF TSC2

Figure 4e

0 2 6 8 T 0 2 6 8 T IR 15Gy + Taxol

Supplementary Figure 17:

Figure 4d

Supplementary Figure 4c

Supplementary Figure 4g

Supplementary Figure 18:

Weel1 MK1775 0.2 μM.

1 2 3 4 5 6 7 8 9
 C 5nM 20nM
 V3 T3 V3R V3 T3 V3R V3 T3 V3R

Supplementary Figure 5g

Supplementary Figure 6g

REVIEWERS' COMMENTS:

Reviewer #1 (Remarks to the Author):

The revised manuscript by Peng and colleagues is significantly improved and addresses most, if not all, of reviewers comments/concerns. The authors performed additional experimentation that strengthen the study. They made the appropriate changes to incorporate these new studies, and to address comment or to clarify ambiguities.

Overall, this remains a very strong study that will have a significant impact in the TSC and LAM field.

Reviewer #2 (Remarks to the Author):

In the revised manuscript, authors provide supporting evidence for the functional connection between mTOR pathway and the G2 checkpoint recovery. In particular, they demonstrate that expression of KDM4B rescues cyclin B levels in mTOR-depleted cells. In addition, they demonstrate that loss of TSC2 caused checkpoint override after exposure to ionizing radiation and resulted in mitotic catastrophe. In the summary, authors have now addressed all my critical comments and in my view the manuscript is now suitable for publication in Nature Communications.

Reviewer #3 (Remarks to the Author):

The authors have addressed my questions regarding assays performed with a reverse phase protein array with adequate information and references. I do not have any further questions.